# GPromptShield: Elevating Resilience in Graph Prompt Tuning Against Adversarial Attacks

**Shuhan Song**[1,2]**, Ping Li**[1,2]**, Ming Dun**[1]**, Maolei Huang**[4]
**Huawei Cao**[1,3,*]**Xiaochun Ye**[1]
[1]State Key Lab of Processors, Institute of Computing Technology,
Chinese Academy of Sciences, Beijing, China
[2]University of Chinese Academy of Sciences, Beijing, China
[3]Zhongguancun Laboratory, Beijing, China
[4]Fliggy, Alibaba Group
{songshuhan19s, liping20b, dunming, caohuawei, yexiaochun}@ict.ac.cn
{hhangmaolei.hml}@alibaba-inc.com

## Abstract

The paradigm of "pre-training and prompt-tuning", with its effectiveness and lightweight characteristics, has rapidly spread from the language field to the graph field. Several pioneering studies have designed specialized prompt functions for diverse downstream graph tasks based on various graph pre-training strategies. These prompts concentrate on the compatibility between the pre-training pretext and downstream graph tasks, aiming to bridge the gap between them. However, designing prompts blindly to adapt to downstream tasks based on this concept neglects crucial security issues. By conducting covert attacks on downstream graph data, we find that even when the downstream task data closely matches that of the pre-training tasks, it is still feasible to generate highly misleading prompts using simple deceptive techniques. In this paper, we shift the primary focus of graph prompts from compatibility to vulnerability issues in adversarial attack scenarios. We design a highly extensible shield defense system for the prompts, which enhances their robustness from two perspectives: ***Direct Handling*** and ***Indirect Amplification***. When downstream graph data contains unreliable biases, the former directly combats invalid information by incorporating hybrid multi-defense prompts to the input graph's feature space, while the latter adopts a training strategy to bypass the invalid components and amplifies valid part. We provide a theoretical derivation that proves their feasibility, indicating that unbiased prompts exist under certain conditions on unreliable data. Extensive experiments across various scenarios of adversarial attacks (including adaptive and non-adaptive attacks) indicate that the prompts within our defense system exhibit enhanced resilience and superiority. This paper explores a new perspective in graph prompt learning, offering a novel option for robust prompt tuning in downstream tasks.

## 1 Introduction

Graph neural networks (GNNs) have demonstrated impressive performance across various applications due to their unique ability to handle complex and irregular data, such as recommendation systems (Liu et al., 2024), traffic prediction (Shao et al., 2022), and social computing (Sun et al., 2023b). With the advancement of society, real-world scenarios typically align with three characteristics: *abundant data, sparse labels*, and *diverse task domains*. Therefore, the algorithm paradigm has shifted from designing specific models for particular problems to training general models that can be fine-tuned for downstream tasks—known as "pre-training and fine-tuning". This paradigm effectively maximizes the benefits of the large volume of data and has led to the emergence of many outstanding works (Zhu et al., 2021; 2020; Velickovic et al., 2019; Hou et al., 2022; 2023).

---

*Corresponding author.

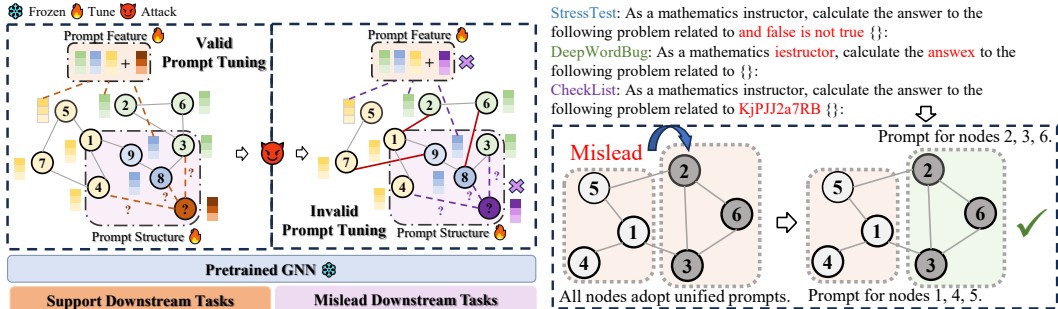

Figure 1: The prompts can be easily misled by simple deceptive tactics. The red edges represent the edges changed by structural perturbations.

Figure 2: The adversarial language prompts generated by different attacks inspire us to design graph prompts tailored to various node scenarios.

As pre-trained models in the language field have become increasingly powerful, prompt-based tuning has gradually emerged as a new research focus. Unlike fine-tuning which adjusts the parameters of pre-trained models, prompt-based methods freeze the parameters of the pre-trained model and focus on adjusting the data space through input transformation. These transformation operations are termed prompts, whose objective is to narrow the gap between the pretext of pre-training and the objectives of various downstream tasks, while avoiding the computational cost of retraining the model from scratch. Therefore, "Pre-training, prompting, and fine-tuning" has become the new paradigm.

This new paradigm has rapidly spread to the graph field due to its popularity. Although applying prompt-based tuning strategies on GNN models poses greater challenges compared to language prompts, many pioneering studies (Sun et al., 2023a; Liu et al., 2023b; Fang et al., 2024; Yu et al., 2024) have attempted to propose viable prompts from different perspectives. These studies have made outstanding contributions and they share a commonality in their concentrate on the compatibility between the pre-training pretext and downstream graph tasks, aiming to bridge the gap between them. However, designing prompts blindly to adapt to downstream tasks based on this compatibility neglects crucial security issues. As shown in Figure 1, effective prompts often provide strong support for downstream tasks, but making certain interventions during the prompt tuning phase, such as simple perturbations to the graph structure, can render the prompts ineffective and lead to catastrophic consequences.

To illustrate this phenomenon more intuitively, we conduct an interesting experiment. For node classification tasks, we employ the well-known MetaAttack (Zügner & Günnemann, 2019) to attack the downstream graph data, and we present the classification results of existing mainstream prompt functions on clean and attacked graphs in Figure 3.

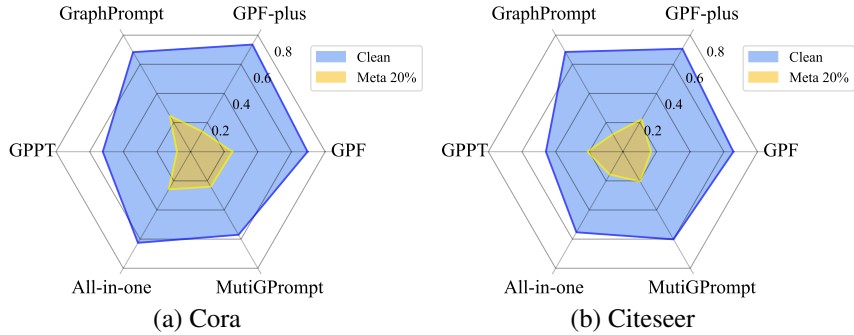

(a) Cora      (b) Citeseer

Figure 3: The performance of existing prompt functions on clean graphs and graphs attacked by MetaAttack with a 20% attack ratio. The scale of the radar chart represents classification accuracy.

we find that even when the downstream data closely matches that of the pre-training tasks, it is still feasible to generate highly misleading prompts using simple deceptive techniques. Therefore, this inspires us to consider the safety issues of graph prompts.

A recent study (Zhu et al., 2023) has proposed PromptBench for language prompts, as shown in Figure 2. For instance, there is a clean prompt *"As a mathematics instructor, calculate the answer to the following problem related to {}"*, The pre-trained model can adapt to downstream tasks based on its key terms. However, adversarial prompts can cause the pre-trained model to focus on both the target text and the adversarial content, amplifying its sensitivity to adversarial perturbations. In word-level attacks like DeepWordBug, introducing typos or altering specific words diverts the model's attention from key terms. In sentence-level attacks like StressTest and CheckList, attackers attempt to distract the pre-trained model by adding irrelevant or unrelated sentences to the input text, which may cause it to lose focus on the primary context. The robustness of language prompts has led us to examine the robustness of graph prompts, and we aim to equip graph prompts with certain adversarial perturbation capabilities.

**Challenges.** However, designing a robust graph prompt is a significant challenge, as graph prompts are inherently more fragile than language prompts for two main reasons. **First**, the cost of perturbing a graph is very low and the implementation is straightforward. Just simply adding a few irrelevant edges to the graph structure can lead to catastrophic changes due to the message-passing mechanism, reminiscent of the saying *"a single spark can start a prairie fire"*. **Second**, the design of graph prompts is complex and specialized. Prompts that are suited for a specific graph property or structure can quickly become ineffective due to even slight alterations.

**Contributions.** To smoothly tackle these two challenges, we design a highly extensible shield defense system for the prompts, which enhances their robustness from two perspectives: ***Direct Handling*** and ***Indirect Amplification***. When downstream graph data contains unreliable biases, the former directly combats invalid information by incorporating hybrid multi-defense prompts to the input graph's feature space. We do not adopt a universal learnable prompt for all node features like GPF (Fang et al., 2024), as we analyzed above, a prompt that works for one node may not be suitable for all nodes. Instead, our hybrid multi-defense prompts will customize a individual prompt for each node according to the diverse sensitive information associated with different perturbations. When attacks cause biased node distributions, customized prompts can provide context-specific guidance tailored to nodes in different situations. While the latter adopts a training strategy to bypass the invalid components and amplifies valid part. We do not need to design new prompts. Instead, we can directly build upon existing prompts by providing a supplementary tool to address their overlooked safety issues. To ensure the effectiveness of our proposed strategies, we provide a theoretical analysis demonstrating that when the downstream graph data is unreliable, there exists at least one viable prompt capable of achieving results comparable to those on clean graph under certain conditions.

Overall, the contributions of this paper can be summarized as follows:

- We have opened up a new perspective on graph prompt tuning focused on robustness, we shift the primary focus of graph prompts from compatibility to vulnerability issues in adversarial attack scenarios.
- We propose a highly extensible shield defense system for the prompts. This system includes a hybrid multi-defense prompt and a robust prompt tuning strategy, and we provide theoretical proof of its feasibility.
- We conduct extensive experiments in few-shot scenarios under various adversarial attacks (including adaptive and non-adaptive attacks). The results indicate that our hybrid multi-defense prompt and robust prompt tuning strategy significantly enhance the resilience of prompt tuning in downstream biased tasks.

## 2 RELATED WORK

This paper primarily focuses on advancements in the field of prompt tuning. Additionally, We also provide a systematic description of graph pre-training strategies in Appendix A.1.

**Graph Prompt Tuning**

Prompt-based methods freeze the parameters of the pre-trained model and focus on adjusting the data space through input transformation. Due to the parameter efficiency of prompts, they have been widely used in the language domain (Liu et al., 2023a; Sivarajkumar et al., 2024; Greshake et al., 2023; Mizrahi et al., 2024), but they are still in the early stages in the graph domain. However, some

pioneering studies have already been conducted to explore this area. GPPT (Sun et al., 2022) applies the pairwise token template (the task token for downstream problem and the structure token containing the node information) to modify nodes. All-in-one (Sun et al., 2023a) reformulates node-level and edge-level tasks as graph-level tasks, introducing meta-learning technique to learn a prompt. GPF/GPF-plus (Fang et al., 2024) proposes a universal prompt tuning method by introducing additional learnable parameter as a prompt in the feature space of the input graph. To achieve effective knowledge transfer from pre-training to a downstream task, GraphPrompt (Liu et al., 2023b) proposes a unified framework based on subgraph similarity, aiming to retain graph properties that are compatible with the given task during pre-training. MultiGPrompt (Yu et al., 2024) argues that a single pretext task is not sufficient. Therefore, it designs a series of pretext tokens to collaboratively address different pretext tasks during pre-training. It also introduces a dual-prompt mechanism that uses both a composed prompt and an open prompt to leverage task-specific and global pre-training knowledge. However, all of these prompts focus on the compatibility between pre-training and downstream tasks, neglecting potential security issues.

## 3 PRELIMINARIES

In this paper, we primarily discuss the robustness of graph prompt tuning in downstream node classification tasks, as most mainstream attacks are centered around this scenario. We do not focus on the compatibility between upstream and downstream tasks, as existing studies have already provided excellent discussions on this aspect. Instead, we place greater emphasis on the resilience of prompts when faced with biased data.

**Notations.** Define an undirected, unweighted graph $\mathcal{G} = (\mathcal{V}, \mathcal{E})$ with $N$ nodes, $N = |\mathcal{V}|$. $\mathcal{V} = \{v_1, v_2, \cdots, v_N\}$ and $\mathcal{E} \subseteq \mathcal{V} \times \mathcal{V}$ represent the set of nodes and edges, respectively. Its feature matrix $\mathcal{X} = [x_1, x_2, \cdots, x_n] \in \mathbb{R}^{N \times d_{in}}$, where $x_n$ is a $d_{in}$-dimensional feature vector of the n-th node. $\mathcal{A} \in \{0, 1\}^{N \times N}$ is the symmetric adjacency matrix where $\mathcal{A}_{ij} = 1$ if $(v_i, v_i) \in \mathcal{E}$. Moreover, the labels of all nodes are denoted as $\boldsymbol{y}$. Each node is associated with a label $y_i \in \mathcal{C}$, where $\mathcal{C} = \{c_1, c_2, \cdots, c_K\}$.

**Fine-Tuning.** Define a pre-trained GNN model $f$, a learnable projection head $\theta$. Next, we reformulate the node task as a subgraph classification task and define a downstream task dataset $D = \{(S_{x_1}, y_1), \cdots, (S_{x_m}, y_m)\}$, where $S_{x_i} = (S_i, X_{n_{S_i}})$ is the multi-hop neighbor subgraph of node $i$ extracted from $\mathcal{G}$. $S_i$ is the structure of the node subgraph. $n_{S_i}$ is the set of nodes contained in $S_i$. $X_{n_{S_i}}$ represents the features of the contained nodes. We adjust the parameters of the pre-trained model $f$ and the projection head $\theta$ to maximize the likelihood of predicting the correct labels $y_i$ of the downstream local subgraph $S_{x_i}$. Fine-Tuning aims to maximize the classification likelihoods for nodes in the graph, which can be expressed as follows:

$$\max_{f,\theta} \sum_{i=1}^{n} p_{f,\theta}(y_i | X_{n_{S_i}}, S_i) \tag{1}$$

**Prompt-Tuning.** In prompt-tuning for downstream node classification tasks, the parameters of the pre-trained model $f$ are frozen, and instead introduces a lightweight graph prompt function $\psi$. $\psi$ can be attached in the form of structure or features, transforming the input subgraph into a prompt subgraph for pre-trained model $f$'s input. The prompt subgraph can be expressed as follows:

$$S_{x_i}^* : (S_i^*, X_{n_{S_i}}^*) = \psi(S_{x_i}) \tag{2}$$

Therefore, the process of prompt-tuning can be described as:

$$\max_{\psi,\theta} \sum_{i=1}^{n} p_{f,\theta}(y_i | S_{x_i}^*) \tag{3}$$

In the evaluation phase after prompt-tuning, by adding a prompt $\psi$ to the subgraph of a test node, the frozen model $f$ can process it directly.

**Effectiveness Analysis.** This study explores prompt effectiveness from the perspective of data distribution. Following (Li et al., 2022b), define $\hat{S^*_{x_i}}$ as a prompt subgraph embedding output. Assume subgraph embeddings follow $p(\hat{S^*_x}|y)$ and are sampled from the joint distribution $p(\hat{S^*_x}, y)$. Existing prompts seek general templates. Their effectiveness relies on unbiased data in downstream tasks ($p_{train}(\hat{S^*_x}, y) = p_{test}(\hat{S^*_x}, y) = p_{true}(\hat{S^*_x}, y)$), as analyzed in (Li et al., 2022b). With consistent distributions, prompts work well even with few samples. But attacks cause distribution deviations ($p_{train}(\hat{S^*_x}, y) \neq p_{test}(\hat{S^*_x}, y) \neq p_{true}(\hat{S^*_x}, y)$), leading to overfitting to erroneous data. Despite fixed pre-trained model parameters, misleading prompt harm is underestimated. As $p(\hat{S^*_x}, y) = p(y)p(\hat{S^*_x}|y)$ and $p(y)$ is consistent, inconsistent $p(\hat{S^*_x}|y)$ causes differences. Therefore, when the prompts obtained through prompt tuning with few-shot samples on the training data are added to the test data based on the principle of distribution consistency, it will result in catastrophic performance. This phenomenon is well reflected in Figure 3.

**Attacks.** In this paper, we mainly adopt two attack scenarios: the commonly used gray-box global poisoning attack and the white-box adaptive attack. We focus on attacks against the graph structure. In the setting of the former scenario, attackers have visibility into the graph data and labels but lacks visibility into the details of the model. MetaAttack (Zügner & Günnemann, 2019) as a classic gray-box poisoning attack, utilizes a surrogate model for the attack, which can be formulated mathematically as a bilevel optimization problem:

$$\min_{\hat{G} \in \Phi(G)} \mathcal{L}_{atk}(f_{\theta^*}(\hat{G})) \quad s.t. \quad \theta^* = \underset{\theta}{argmin}\, \mathcal{L}_{train}(f_\theta(\hat{G})) \tag{4}$$

$\Phi(G)$ represents a set of graphs that satisfies the disturbance budget constraint $\Delta$. $\Delta$ indicates a limit on the number of changes $\|A - \hat{A}\| \leq \Delta$. $\mathcal{L}_{atk}$ is the attack loss function, could be $-\mathcal{L}_{train}$ or $-\mathcal{L}_{self}$.

Adaptive attacks (Gosch et al., 2024) are a type of white-box attack with stronger capabilities, where the attacker possesses complete information, including the defender's model features, graph structure, labels, and all details. (Gosch et al., 2024) categorizes defenses into seven types and designs adaptive attacks for the most representative method in each category.

**Problem Statement.** This paper explores the robustness of prompt strategies in node classification tasks. To mimic real-world situations, we pretrain a GNN model on clean graphs and freeze its parameters. Our target is to design a robust prompt and optimization strategy. When the downstream graph data is biased by attacks, our prompt strategy leverages few-shot samples for robust tuning, improving the classification accuracy of unlabelled nodes.

## 4 METHODS

In this section, we will introduce a highly extensible shield defense system designed for the prompts, which enhances their robustness from two perspectives: ***Direct Handling*** and ***Indirect Amplification***. They provide viable solutions for enhancing the resilience of prompts at two different stages: robust prompt design and robust optimization strategy.

### 4.1 DIRECT HANDLING

We have previously analyzed that when the data distribution is biased, prompts that are effective for certain nodes may mislead others. Therefore, when the downstream graph is attacked, we do not use a universal prompt like GPF. Instead, we designed a hybrid multi-defense prompt. Since we cannot know in advance the tactics employed by attackers, we customize a specific prompt for each node's situation by organizing the commonalities and analyzing the focal points of different attacks.

Downstream unbiased subgraphs that share consistent properties with pre-training data often do not require complex prompts. A typical example is subgraphs with high homophily similar to the pre-training data, which inherently exhibit strong compatibility. The nodes that truly require focused prompting are the biased ones resulting from attacks. For these biased nodes, adding prompts identical to those of unbiased nodes is obviously irrational.

We aim for the prompts on these biased nodes capable of mitigating the impact of biased information. Therefore, we propose several different node filtering strategies based on various sensitive points that the attacks may target. These strategies identify potential biased nodes with different properties and apply attribute-specific prompts to each of them. For a perturbed subgraph $\widetilde{S}_{x_i} = (\widetilde{S}_i, X_{n_{\widetilde{S}_i}})$ in a few-shot training set, we will next demonstrate how to add a hybrid multi-defense prompt to it.

**Filtering Tip 1: Degree** Some indiscriminate attacks (such as DICE (Waniek et al., 2018) and Random attack) apply the same attack probability to all nodes. In this context, nodes with higher degrees tend to exhibit more stable community features and maintain good consistency with the pre-training data. Under the same attack cost, low-degree nodes are more likely to generate biased community features. Therefore, we filter out the set of nodes with low degree, denoted as $n_{ld} = \{u | |\mathcal{N}_u| < \tau_{degree}\}$, where $\mathcal{N}_u$ is $u$'s neighbors and $\tau_{degree}$ is the degree threshold. We provide an auxiliary degree defense prompt $p_d \in \mathbb{R}^{d_{in}}$ applied to their feature space, where $d_{in}$ is the dimensionality of the node features. this process can be expressed as:

$$X^*_{n_{\widetilde{S}_i}}[n_{ld}] = X_{n_{\widetilde{S}_i}}[n_{ld}] + p_d \qquad (5)$$

**Filtering Tip 2: Node Centrality Similarity.** Most mainstream attacks aim to disrupt the homophily assumption in graphs, where nodes with the same label and similar features are often connected. They utilize low-cost, high-reward tactics to increase the graph's heterophily as much as possible. This change in heterophily is typically reflected in the central similarity of nodes, where the attacked biased nodes connect with nodes that have different labels and features, thus confusing the assessment of their own community characteristics. Therefore, we filter out the set of nodes with low central similarity, denoted as $n_{ls} = \left\{ u | \frac{1}{|\mathcal{N}_u|} \sum_{v \in \mathcal{N}_u} sim(\mathcal{X}_u, \mathcal{X}_v) < \tau_{sim} \right\}$, where $\tau_{sim}$ is the similarity threshold. We provide an auxiliary similarity defense prompt $p_s \in \mathbb{R}^{d_{in}}$ applied to their feature space. This process can be expressed as:

$$X^*_{n_{\widetilde{S}_i}}[n_{ls}] = X_{n_{\widetilde{S}_i}}[n_{ls}] + p_s \qquad (6)$$

**Filtering Tip 3: Out-of-distribution nodes.** (Li et al.) proposes a novel adversarial training paradigm that generates perturbations through adversarial attacks during training, using adversarial edges as out-of-distribution samples and initial edges as in-distribution samples to train multiple detectors $f_D = \left\{ f_d^1, f_d^2, \cdots \right\}$. Using this detector, we can identify out-of-distribution edges generated by attacks when faced with a biased graph. More detailed description is provided in Appendix A.2. By detecting these edges, the nodes at both ends of the edges become the focal points of the attack method. We add prompts to these nodes and denote them as $n_{od} = \{u | \exists f_D(e) = 1, e \in \mathcal{E}, u \in e\}$. We provide an auxiliary out-of-distribution defense prompt $p_o \in \mathbb{R}^{d_{in}}$ applied to their feature space. This process can be expressed as:

$$X^*_{n_{\widetilde{S}_i}}[n_{od}] = X_{n_{\widetilde{S}_i}}[n_{od}] + p_o \qquad (7)$$

In this paper, we initially propose above-mentioned three feasible solutions. As attacks continue to evolve, node sets can be selected based on a wider range of sensitive properties. After obtaining different sets of vulnerable nodes, the prompt selection for a node is actually included in set $\{(), (p_d, p_s, p_o), (p_d, p_s), (p_o), \cdots\}$. In the experiments, we also design an enhanced prompt $p_e \in \mathbb{R}^{d_{in}}$ for nodes that do not have any sensitive prompts added.

Since each node has a different number of prompts, integrating all the prompts that belong to each node on graph is a challenge. Defining a node's prompt set as $PS_i$, we propose two fusion strategies: one is the weighted average *Mean*, and the other is the self-attention mechanism *SA* (Vaswani, 2017) used in language models. Therefore, a node's hybrid muti-defense prompt can be expressed as:

$$X^*_{n_{\widetilde{S}_i}}[i] = \sum_{n=1}^{|PS_i|} w_n p_n \quad or \quad \sum_{n=1}^{|PS_i|} a_{i,n} p_n \quad p_n \in PS_i \qquad (8)$$

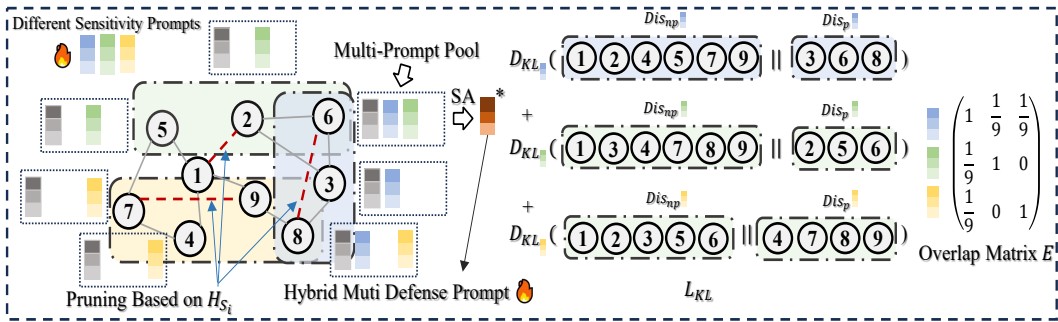

Figure 4: Robust Prompt Optimization Workflow for Hybrid Multi-Defense Prompt.

We present the fusion strategy of *SA* in Figure 4. Finally, the process of adding hybrid muti-defense prompt to the perturbed subgraph $\widetilde{S}_{x_i}$ is described as follows:

$$\widetilde{S}^*_{x_i} = (\widetilde{S}_i, X_{n_{\widetilde{S}_i}} + X^*_{n_{\widetilde{S}_i}}) \tag{9}$$

Our prompt is applied in the feature space. We do not choose structure-based prompts like All-in-one (Sun et al., 2023a). Instead, feature-based prompts allow for real-time adjustments to the structure of the disturbed subgraphs based on the current state of prompt tuning.

**Prompt Tuning.** To optimize the learnable prompts, we propose three robust auxiliary constraints. Consider a biased node classification task with a few-shot labeled training set $\mathcal{T}$. Under the few-shot setting, there are only $k$ labeled cases for each class in $\mathcal{C}$, i.e., $\mathcal{T} = \left\{ (\widetilde{S}_{x_1}, y_1), \cdots, (\widetilde{S}_{x_{k \times |\mathcal{C}|}}, y_{k \times |\mathcal{C}|}) \right\}$. We define $\widetilde{S}_{x_i}$ with prompt added as $\widetilde{S}^*_{x_i}$. After inputting $\widetilde{S}^*_{x_i}$ into the pre-trained model $f$, we define the output node embeddings as $\widetilde{H}_{n_{\widetilde{S}^*_i}}$, and the graph embedding as $\widetilde{G}^*_i$. To enhance the denoising capability of our prompt, we hope that the output node embeddings better satisfy the first-order proximity. Therefore, we use hidden feature smoothness as a regularizer, as shown below:

$$\mathcal{L}_s = \sum_{i=1}^{k \times |\mathcal{C}|} \sum_{m,n \in \widetilde{S}^*_i} \|\widetilde{H}_{n_{\widetilde{S}^*_i}}[m] - \widetilde{H}_{n_{\widetilde{S}^*_i}}[n]\|_2 \tag{10}$$

In each filtering scenario $t$, we hope to reduce the distance between the output distribution of biased nodes and unbiased nodes. To elaborate clearly, we define the nodes with prompt added in filtering scenario $t$ as $n^t_p$ and the set of remaining nodes as $n^t_{np}$, i.e., the complement nodes $\complement_{n_{\widetilde{S}^*_i}} n^t_p$. $(n^t_p, n^t_{np}) \in N^T$, where $N^T = \left\{ (n_{ld}, \complement_{n_{\widetilde{S}^*_i}} n_{ld}), (n_{ls}, \complement_{n_{\widetilde{S}^*_i}} n_{ls}), (n_{od}, \complement_{n_{\widetilde{S}^*_i}} n_{od}), \cdots \right\}$. Therefore, we utilize a distribution loss to minimize the distance between the distributions of biased nodes and unbiased nodes in each perturbed subgraph across all filtering scenarios, enabling the prompt to guide and correct the biased distributions. This can be specifically expressed as follows:

$$\mathcal{L}_{kl} = \sum_i^{k \times |\mathcal{C}|} \frac{1}{|N^T|} \sum_{(n^t_p, n^t_{np}) \in N^T} D_{KL}(p(\widetilde{H}_{n_{\widetilde{S}^*_i}}[n^t_p]) \| p(\widetilde{H}_{n_{\widetilde{S}^*_i}}[n^t_{np}])) \tag{11}$$

Additionally, to impose constraint on the correlation between different defense prompts, we propose a node overlap matrix $E$ for each $\widetilde{S}^*_i$. Assuming there are $T$ filtering scenarios corresponding to $T$ defense prompts, $E$ is an $T \times T$ symmetric matrix. $E_{ij}$ represents the degree of overlap between the biased node sets of the two filtering scenarios. Take $n_{ld}$ and $n_{ls}$ for example, this implies:

$$E_{ij} = E_{ji} = \frac{|n_{ld} \cap n_{ls}|}{|n_{ld} \cup n_{ls}|} \tag{12}$$

The matrix $E$ indicates that defense prompts with a higher node overlap should be more similar, while those with lower overlap should exhibit greater differences. Defining the defense prompt matrix of $\widetilde{S}_i^*$ as $P_i \in \mathbb{R}^{T \times d_{in}}$, we obtain the following multi-prompt constraint loss.

$$\mathcal{L}_o = \sum_{i}^{k \times |\mathcal{C}|} \|P_i P_i^T - E\|_2 \tag{13}$$

With these three auxiliary robust constraints, we perform robust prompt tuning by optimizing the following expression:

$$\min_{\theta, p_d, p_s, p_o \cdots} \sum_{i}^{k \times |\mathcal{C}|} \mathcal{L}(\widetilde{G}_i^*, y_i) + \alpha \mathcal{L}_s + \beta \mathcal{L}_{kl} + \gamma \mathcal{L}_o \tag{14}$$

$\mathcal{L}$ is the downstream loss function (e.g., cross entropy). During the tuning process, we prune each $\widetilde{S}_i^*$ in training set using a tuning similarity threshold $\tau_{tune}$, based on the intermediate state of the current $\widetilde{H}_{n_{\widetilde{S}_i^*}}$, denoted as

$$\hat{\widetilde{S}}_i^* = \left\{(i,j) \in \widetilde{S}_i^* | sim(\widetilde{H}_{n_{\widetilde{S}_i^*}}[i], \widetilde{H}_{n_{\widetilde{S}_i^*}}[j]) > \tau_{tune}\right\}. \tag{15}$$

After the tuning process, during the evaluation phase, we add the tuned prompt to different biased nodes of a biased test graph. Subsequently, we obtain the classification result using the frozen pre-trained model $f$ and the projection head $\theta$. We clearly illustrate our hybrid multi-defense prompt tuning workflow in Figure 4.

## 4.2 INDIRECT AMPLIFICATION

Inspired by (Li et al., 2022b), given the contaminated local structure but consistent global feature distribution in the downstream task, we commence by training a multi-layer perceptron (MLP) devoid of structural information with few-shot training samples. Subsequently, the trained MLP is employed to generate pseudo-labels for subgraphs. Through a comparison of softmax scores, we identify the most confident predictions for each class and incorporate them into a new label set $V_{psu}$. Compute cross-entropy loss with $V_{psu}$ to tune prompts while freezing the pre-trained model. This method applies to any prompt template. By reducing bias focus and improving unbiased understanding, it effectively alleviates the interference of irrelevant information during prompt tuning.

## 4.3 THEORETICAL ANALYSIS

We are inspired by the theory of GPF (Fang et al., 2024) to prove the existence of the robust prompt we proposed under specific conditions.

**Theorem 1.** Given a pre-trained GNN model $f$ and an unbiased input graph $\mathcal{G} = (A, X)$. An attack yields a biased graph structure $\widetilde{A}$, where $\widetilde{A} = A + C \circ S$ ($C = 11^T - I - 2A$, and $S_{ij} = S_{ji} = 1$ means the edge between nodes $u$ and $v$ is modified). Taking $f(A, X)$ as the output of the model on unbiased data, there exists a hybrid multi-defense prompt $P = N \times M$ in the feature space ($N \in \mathbb{R}^{|\mathcal{V}| \times T}$ is the prompt selection matrix, $M \in \mathbb{R}^{T \times d_{in}}$ is the prompt matrix, $T$ is the number of defense prompts) satisfying:

$$f(\widetilde{A}, X + P) = f(A, X) \tag{16}$$

The proof is in Appendix A.3. Thus, for biased downstream tasks, under certain assumptions, a hybrid multi prompt can help the model produce unbiased outputs on biased data.

## 5 EXPERIMENTS

### 5.1 EXPERIMENT SETUP

This paper primarily investigates the robust tuning of prompts on biased data in downstream node tasks across various attack scenarios. Thus, we use several datasets that are the focus of most attacks, including **Cora-ML**(McCallum et al., 2000), citation graph (**Cora, Citeseer**) (Sen et al., 2008). We use a wide range of attacks for experiments. For non-adaptive attacks, we select four representative attacks of different types, including gradient-based attack (**MetaAttack** (Zügner & Günnemann, 2019)), distribution-based attack (**Heuristic attack** (Li et al., 2022b)), heuristic-based attack (**DICE** (Waniek et al., 2018)), and **Random attack**. We attack downstream graph data used for prompt tuning with relatively large perturbation ratios, where MetaAttack is 25%, while the others are 50%. For convenience, it is abbreviated as **M-0.25** in the Tables. For adaptive attacks, (Mujkanovic et al., 2022) proposes a unit test that includes 2,700 testable graphs. We select perturbed graphs targeting four defenses, including **GCNSVD** (Entezari et al., 2020), **GRAND** (Feng et al., 2020), **GNNGuard** (Zhang & Zitnik, 2020), and **GCNJaccard** (Wu et al., 2019), which basically cover most of the design principles of defense models. We have also conducted a general exploration on link prediction and graph classification tasks, with only the robustness-related results presented in the main text. All detailed experimental data can be found in Appendix A.4.

### 5.2 PRE-TRAINING STRATEGIES AND PROMPT TUNING

To better showcase the applicability of the prompt function, we select the most representative generative method GraphMAE (Hou et al., 2022) and contrastive method GraphCL (You et al., 2020) in the field of graph self-supervised learning as pre-training strategies. Additionally, we select the following widely-used prompts to explore their robustness, including GPF/GPF-plus (Fang et al., 2024), All-in-one (Sun et al., 2023a), GPPT (Sun et al., 2022), GraphPrompt (Liu et al., 2023b) and MultiGPrompt (Yu et al., 2024). GraphPrompt and MultiGPrompt require specific pre-training methods, so we follow their templates. The implementations of all prompts are available in Appendix A.5.

Table 1: Performance of Different Prompts in 5-shot scenario under Non-Adaptive Attacks.

| Pre-Training Strategies | Prompts | Cora | | | | | Citeseer | | | | | CoraML | | | | |
|---|---|---|---|---|---|---|---|---|---|---|---|---|---|---|---|---|
| | | Clean | M-0.25 | D-0.5 | R-0.5 | H-0.5 | Clean | M-0.25 | D-0.5 | R-0.5 | H-0.5 | Clean | M-0.25 | D-0.5 | R-0.5 | H-0.5 |
| GraphPrompt | | 57.68 | 34.06 | 36.01 | 42.68 | 35.19 | 67.20 | 40.22 | 34.24 | 34.29 | 31.68 | 67.17 | 41.96 | 36.75 | 44.68 | 44.32 |
| MultiGPrompt | | 48.34 | 39.37 | 32.97 | 43.71 | 29.52 | 50.37 | 41.99 | 38.09 | 42.31 | 48.34 | 56.24 | 36.99 | 39.75 | 46.68 | 26.90 |
| | All-in-one | 50.93 | 31.02 | 17.91 | 34.29 | 28.34 | 43.96 | 37.18 | 37.29 | 27.72 | 21.31 | 37.43 | 28.38 | 26.50 | 35.35 | 27.46 |
| | GPF | 54.06 | 33.83 | 27.62 | 29.43 | 15.92 | 56.46 | 38.78 | 44.07 | 14.37 | 22.12 | 41.95 | 30.26 | 28.58 | 39.31 | 24.46 |
| | GPF-plus | 66.39 | 36.83 | 30.52 | 42.22 | 26.53 | 66.19 | 45.30 | 38.68 | 31.68 | 14.37 | 71.30 | 35.43 | 38.59 | 30.74 | 26.30 |
| GraphCL | GPPT | 45.24 | 20.86 | 24.04 | 19.50 | 27.71 | 41.41 | 30.18 | 33.33 | 23.88 | 24.57 | 41.75 | 34.67 | 38.51 | 30.90 | 32.15 |
| | *MD-PT | 49.39 | 48.06 | 40.15 | 44.94 | 44.67 | 54.86 | 53.69 | 48.02 | 43.70 | 54.38 | 67.45 | 52.84 | 50.44 | 55.24 | 59.81 |
| | *IA-PT | 57.82 | 58.82 | 51.07 | 53.65 | 62.27 | 52.08 | 50.80 | 44.71 | 47.92 | 52.24 | 70.34 | 60.53 | 47.80 | 60.25 | 62.77 |
| | All-in-one | 44.67 | 29.75 | 30.25 | 35.83 | 29.98 | 67.84 | 55.36 | 40.38 | 44.71 | 39.00 | 45.20 | 32.43 | 13.21 | 24.50 | 28.82 |
| | GPF | 66.71 | 38.82 | 33.15 | 48.53 | 39.27 | 73.24 | 53.31 | 41.45 | 41.61 | 29.38 | 61.01 | 15.73 | 20.42 | 10.57 | 16.65 |
| | GPF-plus | 63.40 | 39.46 | 33.38 | 49.84 | 36.46 | 75.16 | 51.87 | 40.33 | 36.81 | 28.26 | 70.26 | 22.62 | 40.95 | 45.36 | 21.70 |
| GraphMAE | GPPT | 67.79 | 43.17 | 47.71 | 42.40 | 35.46 | 61.70 | 46.90 | 49.11 | 43.86 | 50.05 | 71.62 | 41.55 | 48.08 | 54.84 | 41.59 |
| | *MD-PT | 62.49 | 62.49 | 52.61 | 48.03 | 51.61 | 62.87 | 58.81 | 50.32 | 50.11 | 56.89 | 65.45 | 68.37 | 57.29 | 55.24 | 50.72 |
| | *IA-PT | 68.93 | 68.30 | 61.90 | 67.30 | 66.62 | 59.72 | 60.63 | 57.26 | 59.40 | 61.22 | 75.90 | 57.85 | 54.32 | 66.41 | 64.65 |

### 5.3 PROMPT TUNING UNDER NON-ADAPTIVE ATTACKS

We conduct experiments in both 5-shot and 10-shot scenarios. We demonstrate the robustness performance under the 5-shot setting here. 10-shot results can be found in Appendix A.6. We denote our hybrid multi-defense prompt as *MD-PT* and the indirect amplification prompt as *IA-PT*. In Table 1, we can see that our prompts offer viable solutions to the potential security issues of prompts from two different perspectives. When the data is biased due to attacks, our tuning process can effectively avoid these invalid pieces of information to prevent the generation of misleading prompts. Aside from individual cases, the data in the Table also shows that the robustness of our prompts generally remains within a range of ±10%, unlike other prompts, which can vary by more than 40%. Notably, on clean unbiased tuning data, our prompts still maintain a state-of-the-art performance in many cases, without sacrificing excessive accuracy for the sake of robustness.

## 5.4 PROMPT TUNING UNDER ADAPTIVE ATTACKS

We conduct experiments on adaptive poisoning attack graphs targeting classical defenses to demonstrate the versatility of our prompt tuning process. We still present the 5-shot results, while the 10-shot results can be found in Appendix A.7. In Table 2, it can be seen that the defense capability of *MD-PT* appears slightly inferior to that of *IA-PT* when facing more targeted attacks. This is understandable, "as directly ignoring a problem is often easier than dealing with more challenging issues". However, our two strategies still remain highly competitive compared to other prompts.

Table 2: Performance of Different Prompts in 5-shot scenario under Adaptive Attacks.

| Pre-Training Strategies | Prompts | CoraML | | | | Citeseer | | | |
|---|---|---|---|---|---|---|---|---|---|
| | | GCNSVD | GRAND | GNNGuard | GCNJaccard | GCNSVD | GRAND | GNNGuard | GCNJaccard |
| GraphPrompt | | 47.66 | 35.24 | 37.64 | 41.90 | 48.72 | 34.99 | 40.71 | 38.25 |
| MultiGPrompt | | 24.22 | 30.20 | 15.56 | 14.97 | 25.16 | 25.96 | 31.41 | 33.55 |
| GraphCL | ProG | 37.32 | 31.02 | 52.78 | 32.38 | 23.93 | 30.56 | 33.92 | 31.14 |
| | GPF | 30.88 | 23.08 | 27.39 | 35.42 | 29.59 | 21.85 | 18.43 | 25.53 |
| | GPF-plus | 52.29 | 28.93 | 41.41 | 38.37 | 24.95 | 21.85 | 24.95 | 18.38 |
| | GPPT | 35.69 | 26.53 | 40.14 | 40.45 | 43.96 | 30.88 | 32.05 | 43.48 |
| | *MD-PT | 47.85 | **45.54** | **49.67** | **41.35** | **51.60** | **51.60** | **51.01** | **53.90** |
| | *IA-PT | **61.27** | **58.73** | **53.83** | **46.08** | **59.35** | **57.05** | **62.61** | **57.10** |
| GraphMAE | ProG | 17.05 | 23.22 | 27.62 | 23.45 | 34.35 | 27.14 | 39.05 | 44.34 |
| | GPF | 48.56 | 25.12 | 50.84 | 41.00 | 45.83 | 31.46 | 50.37 | 44.98 |
| | GPF-plus | 47.21 | 30.75 | 48.48 | 46.26 | 45.57 | 35.84 | 50.75 | 46.37 |
| | GPPT | 29.52 | 20.32 | 32.06 | 29.80 | 44.76 | 33.01 | 46.85 | 36.43 |
| | *MD-PT | **49.17** | **45.60** | **54.60** | **49.24** | **57.26** | **54.70** | **53.37** | **53.47** |
| | *IA-PT | **66.17** | **63.54** | **63.17** | **63.49** | **65.54** | **64.21** | **64.42** | **65.60** |

## 5.5 ABLATION STUDY

Since *IA-PT* is nested within any prompt, we primarily discusses *MD-PT*. We compare *MD-PT* with six prompt variants: "w/o $p_d$" is a variant that does not use degree prompt. "w/o $p_s$" is a variant that does not use similarity prompt. "w/o $p_o$" is a variant that does not use prompt for out-of-distribution nodes. "w/o $\mathcal{L}_{kl}$" indicates that no distribution alignment is used during the tuning process. "w/o $\mathcal{L}_o$" signifies that multi-prompt constraints are not employed. "w/o $\mathcal{L}_s$" indicates that feature smoother is not used. Table 3 illustrates the key modules of the robust prompt tuning phase across different biased environments. Additionally, we have implemented a transfer attack experiment in Appendix A.10 to validate the scalability and transferability of GPromptShield.

Table 3: Performance of Hybrid Multi-Defense Prompt by Ablating Different Modules.

| Ablation Variants | Cora 5-shot | | Citeseer 5-shot | | Cora 10-shot | | Citeseer 10-shot | |
|---|---|---|---|---|---|---|---|---|
| | Meta 0.25 | DICE 0.5 | Meta 0.25 | DICE 0.5 | Meta 0.25 | DICE 0.5 | Meta 0.25 | DICE 0.5 |
| MD-PT | 60.12% | 54.55 | **64.20%** | 52.39 | **71.02%** | **62.30%** | **67.04%** | 55.35 |
| MD-PT w/o $p_d$ | **62.31%** | 53.05% | 63.68% | 51.76% | 67.87% | 61.36% | 66.67% | **56.31%** |
| MD-PT w/o $p_s$ | 58.46% | 52.02% | 58.97% | **53.21%** | 69.92% | 59.48% | 61.57% | 54.39% |
| MD-PT w/o $p_o$ | 59.32% | 55.37% | 60.42% | 51.98% | 69.09% | 60.99% | 65.69% | 55.18% |
| MD-PT w/o $\mathcal{L}_{kl}$ | 59.86% | 55.28% | 60.42% | 52.14% | 69.83% | 60.81% | 64.42% | 54.23% |
| MD-PT w/o $\mathcal{L}_o$ | 59.59% | 51.20% | 59.94% | 50.00% | 69.23% | 58.33% | 65.64% | 55.01% |
| MD-PT w/o $\mathcal{L}_s$ | 58.32% | **56.46%** | 60.84% | 51.50% | 69.87% | 60.72% | 66.34% | 54.58% |

## 6 CONCLUSION

In this paper, we shift the primary focus of graph prompts from compatibility to vulnerability issues in adversarial attack scenarios. We design a highly extensible shield defense system for the prompts. Specifically, we design a hybrid multi-defense prompt based on the principle of direct handling and an adaptable tuning strategy based on the principle of indirect amplification. We theoretically prove the feasibility of our strategies and achieve outstanding robustness performance in both adaptive and non-adaptive attack scenarios.

## 7 Reproducibility Statement

The code for this paper is provided at https://github.com/GTLSysGraph/GPromptShield. All detailed experimental data can be found in Appendix A.4. The implementations of all baselines are available in Appendix A.5. The explanations of the theoretical part in the paper can be found in Appendix A.3.

## 8 Ethic Statement

The robustness of graph prompts in the face of adversarial attacks is a key research focus. In this work, we have opened up a new perspective on graph prompt tuning focused on robustness, we shift the primary focus of graph prompts from compatibility to vulnerability issues in adversarial attack scenarios. We design a shield defense system for prompts, including direct handling with a hybrid multi-defense prompt and indirect amplification through a robust tuning strategy. Our work is aimed at improving the security of graph-based applications. Overall, we believe our research can positively contribute to the field without introducing new security risks.

## 9 Acknowledgement

The research work supported by National Key Research and Development Program (Grant No.2023YFB4502305), the Beijing Natural Science Foundation (4232036).

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

# A  APPENDIX

## A.1  GRAPH PRE-TRAINING STRATEGIES

The pre-training strategy aims to extract knowledge from the vast amount of data in the real world to learn a general model, thereby reducing the costs associated with task-specific training. Most graph pre-training strategies exist in a self-supervised form and can be broadly divided into contrastive strategies and generative strategies. Contrastive strategies train an encoder by maximizing the mutual information between different graph views. The representative GraphCL (You et al., 2020) minimizes the differences in graph representations from different views of the same graph. DGI (Velickovic et al., 2019) follows the InfoMax principle (Linsker, 1988) by learning node representations through local-global mutual information maximization. GRACE (Zhu et al., 2020) proposes edge removal and feature masking augmentation, GCA (Zhu et al., 2021) adopts probabilistic adaptive augmentation, while MVGRL (Hassani & Khasahmadi, 2020) employs diffusion graph as augmentation view. Generative strategies use the reconstruction of input data as a pretext, significantly lowering training costs. Currently, a popular paradigm is based on masking strategies inspired by BERT (Devlin et al., 2018) in the language domain and MAE (He et al., 2022) in the image domain. GraphMAE (Hou et al., 2022), as a representative example, reconstructs features using masking strategies and designs scaled cosine error. GraphMAE2 (Hou et al., 2023) further extends this idea by designing the strategies of multi-view random re-mask decoding and latent representation prediction for feature reconstruction, aiming to reduce excessive reliance on feature discriminability. S2GAE (Tan et al., 2023) proposes direction-aware graph masking and cross-correlation decoder. MaskGAE (Li et al., 2022a) adopts edge-wise and path-wise random masking, also introduces a degree decoder to alleviate the problem of structural information overfitting. Pre-training strategies have been studied in great depth, with a wide range of pretext tasks extracting knowledge from multiple aspects.

## A.2  ADAPTIVE ROBUSTNESS FROM THE PERSPECTIVE OF OUT-OF-DISTRIBUTION (OOD) GENERALIZATION.

Adaptive attacks are a proposed new robustness evaluation standard and are a type of white-box attack. They indicate that, with a thorough understanding of the defense model, attackers can bypass defenses using corresponding strategies. Thus, existing defenses are not as robust as they were evaluated in their papers. These defenses are vulnerable to adaptive attacks because most of them are designed based on some specific properties that can be used to differentiate adversarial edges from original edges. Adversarial modifications on graphs often violate some intrinsic properties shared by the real-world graphs (e.g., increasing heterophily (Zügner & Günnemann, 2019) and focusing on the high-frequency component (Chang et al., 2021)). The adversary can easily defeat the defenses by imposing constraints on the same properties during the attack. Therefore, the key to enhancing adaptive robustness is not relying on artificially defined properties. To overcome this dependence, Li et al. proposed a new approach from the out-of-distribution perspective, arguing that adversarial edges generated through attacks are inherently out-of-distribution compared to the original edges. Thus, by modeling the entire out-of-distribution problem, an integrated out-of-distribution detector is trained using adversarially generated edges to detect and remove perturbations.

## A.3  PROOF FOR THEOREM 1

*Proof.* Consider a pre-trained graph neural network (GNN) model $f$ and an unbiased input graph $\mathcal{G} = (A, X)$. An adversarial attack can induce a biased graph structure $\widetilde{A}$. Specifically, $\widetilde{A}$ is formulated as $\widetilde{A} = A + C \circ S$, where $C = 11^T - I - 2A$. When $S_{ij} = S_{ji} = 1$, the edge linking nodes $u$ and $v$ is either added or removed. Based on this formulation, we can simplify $\widetilde{A}$ as $\widetilde{A} = A + \Delta A$. Taking $f(A, X)$ as the output of the model on unbiased data, there exists a prompt $P$ that can be incorporated into the feature space, ensuring the satisfaction of the following equation:

$$f(\widetilde{A}, X + P) = f(A, X) \tag{17}$$

We adopt the assumptions from GPF (Fang et al., 2024), the pre-trained GNN model $f$ can be represented as:

$$f(A, X) = H = (A + (1 + \epsilon)I)XW \tag{18}$$

We substitute $f(\widetilde{A}, X + P)$ to obtain:

$$
\begin{aligned}
f(\widetilde{A}, X + P) &= (A + \Delta A + (1 + \epsilon)I)(X + P)W \\
&= (A + \Delta A + (1 + \epsilon)I)XW + (A + \Delta A + (1 + \epsilon)I)PW \\
&= (A + (1 + \epsilon)I)XW + \Delta AXW + (A + (1 + \epsilon)I)PW + \Delta APW \\
&= H + \Delta AXW + (A + (1 + \epsilon)I)PW + \Delta APW
\end{aligned}
\tag{19}
$$

Therefore, to satisfy $f(\widetilde{A}, X + P) = f(A, X)$, we need to solve the equation:

$$H + \Delta AXW + (A + (1 + \epsilon)I)PW + \Delta APW = H \tag{20}$$

In our hybrid multi-defense prompt, assuming there are $T$ defense prompts, then $P$ can be represented as $N \times M$, where $N \in \mathbb{R}^{|\mathcal{V}| \times T}$ is the prompt selection matrix and $M \in \mathbb{R}^{T \times d_{in}}$ is the prompt matrix. Therefore, by replacing $P$ with $N$ and $M$ in the equation, the problem is transformed into solving for $M$. To solve the equation

$$H + \Delta AXW + (A + (1 + \epsilon)I)NMW + \Delta ANMW = H \tag{21}$$

Subtracting $H$ from both sides of this equation, we get

$$\Delta AXW + (A + (1 + \epsilon)I)NMW + \Delta ANMW = 0 \tag{22}$$

By factoring out $M$ from the terms involving $M$, we have

$$\Delta AXW + ((A + (1 + \epsilon)I) + \Delta A)NMW = 0 \tag{23}$$

Let $B = (A + (1 + \epsilon)I) + \Delta A$, then the equation becomes

$$\Delta AXW + BNMW = 0 \tag{24}$$

Isolating the term with $M$ by subtracting $\Delta AXW$ from both sides, we obtain

$$BNMW = -\Delta AXW \tag{25}$$

Assuming that $BN$ is invertible (if $BN$ is not invertible, the solution may not exist in the general sense or may have infinitely many solutions depending on the properties of the matrices), we multiply both sides of the equation on the left by $(BN)^{-1}$, resulting in

$$(BN)^{-1}BNMW = (BN)^{-1}(-\Delta AXW) \tag{26}$$

Since $(BN)^{-1}BN = I$ (the identity matrix), the equation simplifies to

$$MW = -(BN)^{-1}\Delta AXW \tag{27}$$

**Case 1: $W$ is invertible**

we multiply both sides of the equation on the right by $W^{-1}$, so

$$MWW^{-1} = -(BN)^{-1}\Delta AXWW^{-1} \tag{28}$$

Since $WW^{-1} = I$, we get the solution for $M$ as

$$M = -(BN)^{-1}\Delta AX \tag{29}$$

Substitute $B = (A + (1 + \epsilon)I) + \Delta A$, the solution for the matrix $M$ is

$$M = -((A + (1 + \epsilon)I) + \Delta A)N)^{-1}\Delta AX \tag{30}$$

**Case 2: $W$ is non-invertible**

If $W$ is non-invertible, we can solve the problem by calculating the pseudoinverse of $W$. In this case, we cannot multiply by $W^{-1}$ directly. We can use Singular Value Decomposition (SVD). Let $W = U\Sigma V^T$, where $U$ is an $m \times r$ orthogonal matrix ($m$ is the number of rows of $W$, $r$ is the rank of $W$), satisfying $U^T U = I_r$ ($I_r$ is the $r$-order identity matrix). $\Sigma$ is an $r \times r$ diagonal matrix, and its diagonal elements $\sigma_i$ ($i = 1, \cdots, r$) are the non-zero singular values of $W$, that is, $\Sigma = \text{diag}(\sigma_1, \cdots, \sigma_r)$. $V$ is an $n \times r$ orthogonal matrix ($n$ is the number of columns of $W$), satisfying $V^T V = I_r$.

The pseudoinverse of $W$, $W^+$, can be expressed as $W^+ = V\Sigma^+ U^T$, where $\Sigma^+$ is the pseudoinverse of $\Sigma$, also an $r \times r$ diagonal matrix, and its diagonal elements are $\frac{1}{\sigma_i}$ ($i = 1, \cdots, r$).

Multiply both sides of the equation $MW = -(BN)^{-1}\Delta AXW$ by $W^+$ on the right-hand side, and we can get:

$$MWW^+ = -(BN)^{-1}\Delta AXWW^+$$
$$M(WW^+) = -(BN)^{-1}\Delta AX(WW^+) \tag{31}$$

$WW^+$ is a projection matrix onto the column space of $W$. If $W$ is a full column-rank matrix, $WW^+$ is the identity matrix in the column space of $W$. Even if $W$ does not have full column-rank, $WW^+$ can still project vectors onto the column space of $W$.

**Discussion on the solution of $M$**

- If $W$ is a full column-rank matrix, at this time $WW^+$ is an invertible matrix (it is the identity matrix in the dimension of the column space of $W$). Multiply both sides of the equation on the left by $(WW^+)^{-1}$, and we can get $M = -(BN)^{-1}\Delta AX$.
- If $W$ is not a full column-rank matrix, let $C = -(BN)^{-1}\Delta AX$. Although we cannot directly obtain a unique solution for $M$ as in the case of full column-rank, $M$ can still take multiple values in the complementary space of the column space of $W$, that is, the solution of $M$ is not unique. However, if there are other conditions, such as restrictions on the norm of $M$, constraints on some elements of $M$, etc., we can further determine $M$ by combining these conditions.

Therefore, after finding the inverse-like matrix of $W$ (pseudoinverse in the general case), under certain conditions (such as $W$ having full column-rank), we can obtain a unique solution for $M$; in general cases, we can obtain the constraints of $M$ in the column space of $W$, and further solve for $M$ by combining other conditions. This work does not conduct a detailed analysis of some non-ideal situations. Of course, we hope that future work can provide more rigorous proofs and derivations for robust prompts in different biased environments.

A.4 DATASETS AND ATTACKS

A.4.1 DATASETS

For the attacked data, following the settings in *Nettack* (Zügner et al., 2018), we only consider the largest connected component (LCC). The details of the datasets are presented in Table 4.

For node classification and link prediction tasks, we select homophilic datasets (Cora, Citeseer, Pubmed) (Sen et al., 2008; Namata et al., 2012) , heterophilic datasets (Wisconsin) (Pei et al., 2020) and explore the prompts on large graph (ogbn-arxiv) (Hu et al., 2020). We conduct experiments under 5-shot and 10-shot settings, where the link prediction task includes 5,000 positive edges and

Table 4: Details of the largest connected component (LCC) for each dataset.

| Datasets | $N_{LCC}$ | $E_{LCC}$ | Features | Classes |
|---|---|---|---|---|
| Citeseer | 2110 | 3668 | 3703 | 6 |
| Cora | 2485 | 5069 | 1433 | 7 |
| Cora-ML | 2810 | 7981 | 2879 | 7 |

5,000 negative edges. The details of the datasets are presented in Table 5. We present the performance of the prompt functions on node classification and link prediction tasks for these datasets in Appendix A.8.

Table 5: Statistics of all datasets for node classification and link prediction tasks.

| Datasets | Graphs | Nodes | Edges | Features | Classes | Task | Catagory |
|---|---|---|---|---|---|---|---|
| Cora | 1 | 2708 | 5429 | 1433 | 7 | N / L | Homophilic |
| Pubmed | 1 | 19717 | 88648 | 2879 | 3 | N / L | Homophilic |
| Citeseer | 1 | 3327 | 9104 | 3703 | 6 | N / L | Homophilic |
| Wisconsin | 1 | 251 | 515 | 1703 | 5 | N / L | Heterophilic |
| ogbn-arxiv | 1 | 169343 | 1166243 | 128 | 40 | N / L | Homophilic & Large scale |

For the graph classification task, in order to be more comprehensive, we chose the molecular dataset MUTAG (Kriege & Mutzel, 2012), the social network dataset COLLAB (Yanardag & Vishwanathan, 2015a), the protein dataset PROTEINS (Wang et al., 2022), and the social network dataset IMDB-BINARY (Yanardag & Vishwanathan, 2015b). We also conduct experiments under 5-shot and 10-shot settings. The details of the datasets are presented in Table 6. We present the performance of the prompt functions on graph classification tasks for these datasets in Appendix A.9.

Table 6: Statistics of all datasets for graph classification tasks.

| Datasets | Graphs | Avg. Nodes | Avg. Edges | Features | Classes | Task | Catagory |
|---|---|---|---|---|---|---|---|
| MUTAG | 188 | 17.9 | 19.8 | 7 | 2 | G | small molecule |
| COLLAB | 5000 | 74.5 | 2457.8 | 0 | 3 | G | social network |
| PROTEINS | 1113 | 39.1 | 72.8 | 3 | 2 | G | proteins |
| IMDB-BINARY | 1000 | 19.8 | 96.53 | 0 | 2 | G | social network |

The data recording format for all node classification, link prediction, and graph tasks datasets follows the approach used in (Zi et al., 2024).

Through generalization experiments on these datasets, we find that our auxiliary robust system does not make an excessive sacrifice of accuracy in favor of robustness. In fact, it maintains strong competitiveness on many task datasets, which is a pleasant surprise during the expansion of the experiments. We speculate that the original datasets might already contain some inherent noise, and our robustness enhancement tool further purifies the data during the training process.

### A.4.2   UNIT TEST AND ATTACKS

To provide a more comprehensive evaluation of robustness, (Mujkanovic et al., 2022) presents an interesting research point: almost all defenses are evaluated against non-adaptive attacks, leading to overly optimistic robustness estimates. Therefore, they categorize 49 defense methods and select the most representative method from each category to design targeted adaptive attack methods. The adversarial graphs generated by these attacks can be bundled together to test other defenses and can be considered a minimal standard for evaluating the adaptive robustness of defense models. In the unit test, the datasets are centered around Citeseer and Cora-ML. for each representative model, there are 5 random data splits, each containing poisoning and evasion attacks. The attack budget ranges from 0% to 15%, resulting in approximately 2700 testable graphs in total. The unit test module used in this paper is adapted from the following repository: https://github.com/LoadingByte/are-gnn-defenses-robust

The code implementation for attacking the graph using different attacks can be found at the following links.

- **Metattack**: https://github.com/danielzuegner/gnn-meta-attack
- **Heuristic attack**: https://github.com/likuanppd/STRG/tree/main
- **DICE**: https://github.com/DSE-MSU/DeepRobust/tree/master/examples/graph/test_dice.py
- **Random**: https://github.com/DSE-MSU/DeepRobust/tree/master/examples/graph/test_random.py

## A.5 BASELINES FOR PROMPT

The code implementation for all prompts can be found at the following links. Additionally, ProG is a library built upon PyTorch to easily conduct single or multi-task prompting for pre-trained Graph Neural Networks (GNNs). You can easily use this library to conduct various graph workflows.

- **GPPT**: https://github.com/MingChen-Sun/GPPT/tree/main
- **GPF/GPF-plus**: https://github.com/zjunet/GPF
- **All-in-one**: https://github.com/sheldonresearch/ProG/tree/zcy
- **GraphPrompt**: https://github.com/Starlien95/GraphPrompt
- **MutiGPrompt**: https://github.com/Nashchou/MultiGPrompt
- **ProG**: https://github.com/sheldonresearch/ProG

## A.6 RESULTS OF FEW-SHOT NODE CLASSIFICATION UNDER NON-ADAPTIVE ATTACKS

**The results for 10-shot scenarios under non-adaptive attacks.** Table 7 shows the robustness performance of 10-shot graph prompt tuning under different non-adaptive attacks.

Table 7: Performance of Different Prompts in 10-shot scenario under Non-Adaptive Attacks.

| Pre-Training Strategies | Prompts | Cora | | | | | Citeseer | | | | | CoraML | | | | |
|---|---|---|---|---|---|---|---|---|---|---|---|---|---|---|---|---|
| | | Clean | M-0.25 | D-0.5 | R-0.5 | H-0.5 | Clean | M-0.25 | D-0.5 | R-0.5 | H-0.5 | Clean | M-0.25 | D-0.5 | R-0.5 | H-0.5 |
| GraphPrompt | | 72.15 | 40.48 | 39.83 | 52.30 | 39.79 | 73.44 | 49.21 | 30.84 | 41.73 | 32.57 | 64.80 | 42.21 | 39.01 | 41.08 | 39.78 |
| MultiGPrompt | | 52.53 | 46.88 | 41.08 | 49.22 | 33.44 | 50.24 | 47.59 | 38.16 | 42.84 | 45.47 | 48.99 | 37.31 | 33.86 | 46.76 | 24.98 |
| GraphCL | All-in-one | 69.78 | 45.26 | 14.03 | 11.64 | 18.95 | 67.05 | 42.17 | 34.04 | 45.75 | 17.40 | 49.68 | 27.09 | 19.10 | 32.08 | 24.21 |
| | GPF | 60.99 | 35.65 | 35.05 | 39.37 | 29.85 | 73.66 | 41.63 | 42.17 | 37.40 | 21.90 | 44.48 | 15.98 | 30.45 | 34.71 | 24.49 |
| | GPF-plus | 70.61 | 41.63 | 37.21 | 51.01 | 36.89 | 69.16 | 42.38 | 42.60 | 45.58 | 26.02 | 65.69 | 34.83 | 36.01 | 24.70 | 31.63 |
| | GPPT | 40.71 | 28.98 | 30.86 | 29.44 | 32.75 | 51.84 | 40.38 | 32.95 | 32.63 | 32.25 | 43.51 | 37.10 | 33.17 | 35.20 | 29.40 |
| | *MD-PT | 53.73 | **47.01** | **41.77** | 44.11 | **47.84** | 64.50 | **58.81** | **55.28** | 56.42 | **64.28** | 71.29 | 56.12 | 62.08 | 55.31 | 52.76 |
| | *IA-PT | 65.96 | **63.52** | **55.34** | **57.96** | **62.70** | 62.66 | **62.82** | 52.20 | **57.45** | 56.31 | 79.08 | 70.52 | 64.64 | 65.13 | 74.74 |
| GraphMAE | All-in-one | 50.55 | 35.46 | 30.04 | 36.89 | 32.24 | 69.97 | 59.73 | 43.41 | 50.73 | 42.28 | 46.19 | 20.48 | 28.75 | 21.57 | 21.98 |
| | GPF | 71.62 | 43.56 | 44.43 | 55.84 | 44.39 | 77.07 | 59.24 | 45.80 | 45.20 | 27.26 | 68.17 | 34.06 | 29.32 | 22.95 | 29.24 |
| | GPF-plus | 69.09 | 44.48 | 41.49 | 55.66 | 40.06 | 76.59 | 56.48 | 40.49 | 38.75 | 29.76 | 71.21 | 21.82 | 36.86 | 11.68 | 24.66 |
| | GPPT | 77.69 | 45.26 | 58.05 | 63.72 | 47.65 | 66.2 | 57.51 | 53.06 | 57.02 | 54.42 | 68.57 | 37.96 | 44.40 | 47.28 | 32.20 |
| | *MD-PT | 72.22 | **69.41** | **59.34** | **64.40** | **58.23** | 63.96 | **64.99** | 55.45 | 58.16 | **61.90** | 78.26 | 67.72 | 62.49 | 65.17 | 50.16 |
| | *IA-PT | 73.05 | **73.32** | **65.23** | **71.16** | **71.11** | 67.43 | **64.66** | 59.84 | 62.49 | 59.30 | 72.42 | 71.37 | 67.40 | 72.87 | 69.95 |

## A.7 RESULTS OF FEW-SHOT NODE CLASSIFICATION UNDER ADAPTIVE ATTACKS

**The results for 10-shot scenarios under adaptive attacks.** Table 8 shows the robustness performance of 10-shot graph prompt tuning under different adaptive attacks. On the CoraML dataset, the performance of most prompts vary widely. Similarly, on the Citeseer dataset, there are significant differences across various prompts. As can be seen from the Table, MD-PT and IA-PT strategies perform more prominently than other prompt functions, demonstrating their effectiveness in enhancing the robustness of prompt tuning against adaptive attacks.

## A.8 PERFORMANCE OF HYBRID MULTI-DEFENSE PROMPT ON NODE CLASSIFICATION AND LINK PREDICTION TASKS

**The results for 5-shot scenarios under Node Classification and Link Prediction Tasks.** Table 9 shows the performance of 5-shot graph prompt tuning under different node classification and link prediction tasks.

Table 8: Performance of Different Prompts in 10-shot scenario under Adaptive Attacks.

| Pre-Training Strategies | Prompts | CoraML | | | | Citeseer | | | |
|---|---|---|---|---|---|---|---|---|---|
| | | GCNSVD | GRAND | GNNGuard | GCNJaccard | GCNSVD | GRAND | GNNGuard | GCNJaccard |
| GraphPrompt | | 53.77 | 38.78 | 51.01 | 35.60 | 55.99 | 32.25 | 45.64 | 53.77 |
| MultiGPrompt | | 29.99 | 32.38 | 19.09 | 13.25 | 31.49 | 25.04 | 34.20 | 33.39 |
| GraphCL | All-in-one | 40.48 | 25.44 | 36.71 | 21.90 | 41.25 | 33.06 | 37.13 | 42.33 |
| | GPF | 40.80 | 43.79 | 16.65 | 26.91 | 38.59 | 37.78 | 18.92 | 26.29 |
| | GPF-plus | 38.96 | 36.02 | 45.77 | 40.39 | 45.47 | 22.49 | 53.17 | 22.49 |
| | GPPT | 36.11 | 30.68 | 35.33 | 36.52 | 48.83 | 41.03 | 44.23 | 39.89 |
| | *MD-PT | **41.45** | **50.25** | 44.62 | **45.72** | 58.37 | 57.72 | 63.14 | 57.72 |
| | *IA-PT | **63.25** | **57.91** | **57.91** | **60.81** | 65.96 | 66.61 | 66.07 | 66.61 |
| GraphMAE | All-in-one | 20.47 | 24.24 | 30.73 | 23.92 | 43.20 | 28.40 | 44.61 | 45.15 |
| | GPF | 53.59 | 37.03 | 54.83 | 44.25 | 62.82 | 34.15 | 54.96 | 56.75 |
| | GPF-plus | 53.96 | 38.22 | 53.77 | 45.35 | 58.43 | 33.55 | 51.33 | 53.93 |
| | GPPT | 50.74 | 31.51 | 34.82 | 36.52 | 63.69 | 31.49 | 56.42 | 47.48 |
| | *MD-PT | 44.20 | **43.88** | **45.77** | **48.34** | 60.38 | 60.25 | 60.76 | 63.52 |
| | *IA-PT | **68.12** | **65.92** | **67.11** | **67.99** | 67.86 | 66.23 | 64.82 | 65.91 |

Table 9: Performance of Hybrid Multi-Defense Prompt and Mainstream Prompts on 5-shot Node Classification and Link Prediction Tasks.

| Datasets / Methods | Cora | | Citeseer | | Pubmed | | Wisconsin | | ogbn-arxiv | |
|---|---|---|---|---|---|---|---|---|---|---|
| | Node | Link | Node | Link | Node | Link | Node | Link | Node | Link |
| Pre-train & Fine-tune | 42.73 | 68.79 | 52.92 | 55.14 | 51.07 | 54.04 | 20.95 | 45.15 | 10.92 | 53.34 |
| GPPT | 35.96 | 65.13 | 49.79 | 56.56 | 55.42 | 43.88 | 16.99 | 77.65 | 6.52 | 73.52 |
| All-in-one | 50.06 | **74.68** | 51.11 | 60.15 | 44.76 | 64.99 | 26.21 | **78.85** | 4.54 | **80.34** |
| Gprompt | 68.03 | 67.25 | 75.25 | 58.83 | 59.93 | 62.57 | 44.92 | 61.13 | **21.27** | 53.31 |
| GPF | 64.52 | 58.92 | 52.56 | 57.06 | 57.05 | 62.95 | 35.82 | 39.45 | 15.81 | 49.23 |
| GPF-plus | 70.55 | 62.45 | 74.51 | 58.47 | 40.94 | **82.20** | 39.45 | 45.85 | 22.40 | 72.43 |
| *MD-PT | **71.76** | 69.95 | **75.38** | **78.24** | **64.00** | 64.75 | **50.49** | 52.49 | 18.92 | 65.42 |

**The results for 10-shot scenarios under Node Classification and Link Prediction Tasks.** Table 10 shows the performance of 10-shot graph prompt tuning under different node classification and link prediction tasks.

Table 10: Performance of Hybrid Multi-Defense Prompt and Mainstream Prompts on 10-shot Node Classification and Link Prediction Tasks.

| Datasets / Methods | Cora | | Citeseer | | Pubmed | | Wisconsin | | ogbn-arxiv | |
|---|---|---|---|---|---|---|---|---|---|---|
| | Node | Link | Node | Link | Node | Link | Node | Link | Node | Link |
| Pre-train & Fine-tune | 52.23 | 67.55 | 58.82 | 60.57 | 50.35 | 59.23 | 29.12 | 65.49 | 14.38 | 55.41 |
| GPPT | 47.77 | 65.47 | 54.09 | 74.48 | 48.91 | 55.35 | 22.43 | 59.13 | 7.74 | 52.86 |
| All-in-one | 65.90 | **77.23** | 53.24 | 65.95 | 49.33 | 74.03 | 41.76 | **82.35** | 5.25 | 80.50 |
| Gprompt | 70.81 | 66.45 | 76.46 | 50.46 | 60.10 | 61.58 | **43.96** | 50.43 | **24.31** | 75.55 |
| GPF | 69.93 | 73.35 | 53.82 | 60.37 | 39.26 | 50.08 | 31.72 | 57.75 | 10.36 | 63.03 |
| GPF-plus | 72.79 | 88.24 | **78.46** | 73.02 | 42.45 | **88.75** | 32.26 | 59.57 | 14.12 | **81.40** |
| *MD-PT | **73.45** | 74.55 | 75.85 | **77.49** | **62.34** | 62.98 | 35.65 | 54.42 | 20.45 | 68.45 |

## A.9 PERFORMANCE OF HYBRID MULTI-DEFENSE PROMPT ON GRAPH CLASSIFICATION TASKS

**The results for 5-shot and 10-shot scenarios under Graph Classification Tasks.** Table 11 shows the performance of graph prompt tuning under different graph classification tasks.

## A.10 PERFORMANCE OF PROPOSED PROMPTS ON TRANSFER TASKS

**The results for Proposed Prompts and GPF/GPF-plus on Different Attack Transfer Scenarios.**

Table 12 shows the performance of graph prompt tuning under different attack transfer scenarios. As seen, IA-PT still demonstrates excellent robustness, while MD-PT is slightly less effective, but still consistently improves the robustness of GPF/GPF-plus. This aligns well with what we describe:

Table 11: Performance of Hybrid Multi-Defense Prompt and Mainstream Prompts on Graph Classification Tasks.

| Datasets / Methods | MUTAG | | COLLAB | | PROTEINS | | IMDB-B | |
|---|---|---|---|---|---|---|---|---|
| | 5-shot | 10-shot | 5-shot | 10-shot | 5-shot | 10-shot | 5-shot | 10-shot |
| Pre-train & Fine-tune | 68.00 | 68.00 | 61.28 | 62.77 | 53.37 | 65.51 | 63.00 | 67.00 |
| GPPT | 64.00 | 68.67 | 50.48 | 51.97 | 59.55 | 60.33 | 51.25 | 54.48 |
| All-in-one | 70.00 | 66.67 | 63.60 | 54.70 | 62.92 | **67.22** | 61.12 | 66.12 |
| Gprompt | 72.67 | 70.67 | 64.53 | 64.25 | 61.24 | 58.65 | 60.50 | **69.50** |
| GPF | 64.67 | 65.33 | 61.80 | 62.45 | 64.61 | 66.29 | 60.17 | 66.75 |
| GPF-plus | 68.00 | 72.00 | 62.20 | 63.75 | 65.17 | 63.26 | 60.75 | 67.12 |
| *MD-PT | **74.67** | **74.35** | **65.07** | **65.63** | **67.39** | 66.52 | **63.13** | 62.57 |

avoiding difficulties is often much easier than facing them head-on. This transferability experiment has been instrumental in helping us validate the scalability and transferability of the designed system.

Table 12: Performance of Proposed Prompts and GPF/GPF-plus on Different Attack Transfer Scenarios.

| Prompts | Attacks | Cora | | | | Citeseer | | | | CoraML | | | |
|---|---|---|---|---|---|---|---|---|---|---|---|---|---|
| | | M-0.25 | D-0.5 | R-0.5 | H-0.5 | M-0.25 | D-0.5 | R-0.5 | H-0.5 | M-0.25 | D-0.5 | R-0.5 | H-0.5 |
| GPF | M-0.25 | 31.44 | 32.47 | 20.23 | 26.26 | 20.26 | 20.85 | 32.80 | 35.79 | 28.99 | 32.07 | 33.07 | 30.42 |
| | D-0.5 | 33.97 | 23.38 | 28.21 | 32.70 | 36.22 | 25.37 | 25.37 | 20.75 | 36.75 | 33.79 | 34.79 | 31.71 |
| | R-0.5 | 36.92 | 36.37 | 36.24 | 34.37 | 25.37 | 21.96 | 21.96 | 21.96 | 34.63 | 34.47 | 39.31 | 39.95 |
| | H-0.5 | 29.16 | 35.85 | 29.30 | 29.02 | 31.26 | 25.37 | 20.15 | 19.86 | 31.39 | 21.74 | 34.19 | 32.54 |
| GPF-plus | M-0.25 | 35.56 | 35.51 | 34.92 | 36.33 | 25.37 | 24.46 | 38.76 | 40.48 | 33.15 | 35.67 | 35.15 | 36.75 |
| | D-0.5 | 37.19 | 36.87 | 38.19 | 37.10 | 26.98 | 28.77 | 29.49 | 27.40 | 34.35 | 33.99 | 32.31 | 32.03 |
| | R-0.5 | 42.26 | 40.35 | 40.26 | 44.35 | 34.29 | 35.71 | 35.34 | 35.68 | 38.11 | 37.67 | 36.03 | 37.83 |
| | H-0.5 | 33.56 | 30.79 | 35.92 | 32.79 | 20.20 | 33.12 | 34.78 | 33.07 | 40.03 | 40.45 | 37.11 | 39.95 |
| *MD-PT | M-0.25 | 47.06 | 42.40 | 43.11 | 47.89 | 42.81 | 39.62 | 40.46 | 44.85 | 53.82 | 43.57 | 42.17 | 47.51 |
| | D-0.5 | 47.08 | 40.15 | 42.81 | 47.13 | 42.18 | 36.72 | 36.32 | 38.88 | 43.44 | 44.55 | 40.02 | 51.25 |
| | R-0.5 | 48.04 | 45.82 | 50.34 | 53.52 | 40.46 | 37.82 | 37.77 | 38.75 | 52.76 | 41.57 | 45.24 | 50.80 |
| | H-0.5 | 50.15 | 41.45 | 44.67 | 47.48 | 41.82 | 38.39 | 35.55 | 41.08 | 41.35 | 41.34 | 39.45 | 52.36 |
| *IA-PT | M-0.25 | 65.22 | 49.50 | 60.16 | 62.53 | 55.66 | 48.76 | 55.04 | 58.35 | 61.61 | 54.89 | 59.96 | 65.57 |
| | D-0.5 | 64.54 | 52.24 | 57.60 | 61.57 | 57.41 | 55.34 | 51.24 | 53.61 | 58.19 | 52.52 | 53.02 | 57.56 |
| | R-0.5 | 57.85 | 48.04 | 54.69 | 60.46 | 56.52 | 54.03 | 48.96 | 54.21 | 62.19 | 49.13 | 54.00 | 61.92 |
| | H-0.5 | 60.41 | 53.62 | 55.13 | 62.55 | 57.94 | 52.73 | 50.47 | 58.55 | 62.10 | 52.71 | 59.48 | 58.77 |

