# OpenReview forum: "GPromptShield: Elevating Resilience in Graph Prompt Tuning Against Adversarial Attacks"
_ICLR.cc/2025/Conference — ICLR 2025 Poster_

### Official Review · Reviewer_zGFc · 2024-11-02

**Soundness:** 3
**Presentation:** 3
**Contribution:** 3
**Rating:** 6
**Confidence:** 4

**Summary:**

This paper introduces a robust graph prompt learning method based on adversarial training to defend Graph Neural Networks (GNNs) against adversarial attacks. The method begins by identifying nodes typically targeted in these attacks, such as low-degree nodes, nodes with low central similarity within multi-hop subgraphs, and nodes linked to adversarially perturbed edges. Next, the approach fine-tunes three distinct types of graph prompts, each corresponding to one of the targeted node types. The objective of the prompt tuning process is to ensure: (1) smoothness in node embeddings within multi-hop subgraphs across the three types of graph prompts, (2) consistency between prompt-enhanced and unprompted node embeddings, and (3) smoothness across embeddings from different types of prompts. Finally, the study applies this method to multiple graph prompt learning models, evaluating its defense effectiveness against non-adaptive attacks and its performance in combination with other defense strategies for adaptive and graph poisoning attacks.

**Strengths:**

First of all, this is an interesting work in investigating robust graph prompt tuning to increase the stability of GPL against potential graph poisoning attacks. Especially, if the test graph structure is intentionally manipulated, this method offers a solution to suppress the impact of the poisoning efforts over the decision of GPL.

**Weaknesses:**

There are several points that need to be strengthened:

1/ This work only focuses on node classification-oriented GPL and does not discuss graph poisoning attacks targeting at manipulating node attributes. It is not clear if the proposed robust prompt tuning method can be applied to edge or graph level learning tasks and can be adapted to both node- and edge-manipulation based attacks. I am asking this as GPL is a multi-task system that can transfer knowledge between different types of GNN learning tasks (node, edge and graph classification). From this perspective, the evaluation in this study is not sufficient. Extending the coverage over more graph datasets and GNN learning tasks should be considered. I'd recommend the author refer to the dataset choice and learning task configurations in [1].

[1] All in One: Multi-Task Prompting for Graph Neural Networks, https://arxiv.org/pdf/2307.01504.

2/ The theoretical proof in Section 4.3 is not associated with the main claim (robust prompt tuning) in this work. This theoretical analysis should be dedicated to explain how the robustness is improved via the proposed tuning approach. However, the link between Section 4.3 and the algorithmic design is unclear in the current submission.

3/ The ablation study may also need to consider one situation: what if every node belonging to the three types (low degree, low central similarity and out-of-distribution nodes) has the same graph prompting function ? Besides, which types of the nodes should be given more weights in the robust tuning process, or they are handled equally in adversarial attacks ?

4/ What are out-of-distribution nodes in a graph? Citing the paper (Li et al) is not enough. Further discussion regarding what these nodes are and why they are concerned in adversarial attacks should be involved.

5/ How do you choose the threshold values $\tau_{degree}$ and $\tau_{sim}$ ? Are they set empirically, or is there any protocol to choose adaptively in different GNN adversarial attack scenarios ? Similarly, how do you choose $\tau_{tune}$ in Eq.15 ? Does it mean you completely exclude the subgraphs if the embeddings of nodes in these subgraphs meet the condition in Eq.15? Will it also bring harm to the utility of the trained GPL as it excludes information from the training process.

6/ As an adversarial training method, it is not surprised to see the proposed defensive graph prompts can mitigate one attack method when they are trained to be resilient to this attack. I am wondering if the proposed method has the transferability. In particular, if it is trained with the perturbed graph using one of the attack methods (MetaAttack, Heuristic attack, Random attack, DICE) can be also resilient to the other three attacks.

**Questions:**

Please check the question raised in the comments concerning the weakness of this study.

---

> ### Author Response · Authors · 2024-11-20
> **Thanks to the reviewer and the Response to Point 1**
>
> Dear Reviewer  zGFc,
>
> Thank you very much for the dedicated review of our paper. We are fully aware of the commitment and time your review entails. Your efforts are deeply valued by us.
>
> First and foremost, we sincerely appreciate your comment that our work is interesting, as it makes us feel that you are willing to give us an opportunity to earn your recognition. We highly value each of your suggestions. During this intense period, we have made every effort to conduct numerous additional experiments. In the following responses, we hope to have provided comprehensive supplements and explanations for every point you mentioned.
>
>
>
> > ***POINT 1: Dataset choice and learning task configurations.***
>
> - ***Attack scenario selection.***
>
> In the field of attacks, many methods primarily focus on designing structural attacks on graphs, as feature-based attacks tend to have relatively weaker effects. In contrast, perturbations to the structure result in more significant attack gains. This has been mentioned in many papers like [1,2], which is also why these papers do not specifically address feature-based attacks. Additionally, these well-known attacks mainly focus on node classification tasks. Therefore, our choice of attack scenarios aligns with this approach, making our selection of attack settings both comprehensive and reasonable. Moreover, attacks are often most effective in specific tasks, where the benefits of the attack are maximized. Therefore, exploring robustness in such environments is more meaningful. Certainly, if there are any updated feature-based attacks in the future, we would be glad to further expand on and discuss them in our future work.  : ）!
>
> [1] Adversarial Attacks on Graph Neural Networks via Meta Learning.
>
> [2] Adversarial attacks on neural networks for graph data.
>
> - ***Extension experiments on different task scenarios.***
>
> For additional dataset choices and task settings, we have followed your suggestion and included link prediction and graph classification tasks. We have now completed experiments on all tasks related to graphs！: )
>
> For node classification and link prediction tasks, we select homophilic datasets (Cora,Citeseer,Pubmed) and heterophilic datasets (Wisconsin) and explore the prompts on large graph (ogbn-arxiv). We conduct experiments under 5-shot and 10-shot settings, where the link prediction task includes 5,000 positive edges and 5,000 negative edges.
>
>
> For the graph classification task, in order to be more comprehensive, we chose the molecular dataset MUTAG, the social network dataset COLLAB, the protein dataset PROTEINS, and the social network dataset IMDB-BINARY. We also conduct experiments under 5-shot and 10-shot settings.
>
> Below, we provide a preliminary presentation of the experimental results.
>
> All the datasets information，additional experimental data and more detailed tables will be updated in our revised version shortly.   : )
>
>
>
> ***Performance of Proposed Hybrid Multi-Defense Prompt and Mainstream Prompting Methods on 5-shot Node Classification and Link Prediction Tasks.***
>
> | Methods               | Node (Cora) | Link (Cora) | Node (Citeseer) | Link (Citeseer) | Node (Pubmed) | Link (Pubmed) | Node (Wisconsin) | Link (Wisconsin) | Node (ogbn-arxiv) | Link (ogbn-arxiv) |
> |------------------------|-------------|-------------|-----------------|-----------------|---------------|---------------|------------------|------------------|------------------|------------------|
> | Pre-train & Fine-tune | 42.73       | 68.79       | 52.92           | 55.14           | 51.07         | 54.04         | 20.95            | 45.15            | 10.92            | 53.34            |
> | GPPT                  | 35.96       | 65.13       | 49.79           | 56.56           | 55.42         | 43.88         | 16.99            | 77.65            | 6.52             | 73.52            |
> | ProG                  | 50.06       | **74.68**   | 51.11           | 60.15           | 44.76         | 64.99         | 26.21            | **78.85**        | 4.54             | **80.34**        |
> | Gprompt               | 68.03       | 67.25       | 75.25           | 58.83           | 59.93         | 62.57         | 44.92            | 61.13            | **21.27**        | 53.31            |
> | GPF                   | 64.52       | 58.92       | 52.56           | 57.06           | 57.05         | 62.95         | 35.82            | 39.45            | 15.81            | 49.23            |
> | GPF-plus              | 70.55       | 62.45       | 74.51           | 58.47           | 40.94         | **82.20**     | 39.45            | 45.85            | 22.40            | 72.43            |
> | *MD-PT                | **71.76**   | 69.95       | **75.38**       | **78.24**       | **64.00**     | 64.75         | **50.49**        | 52.49            | 18.92            | 65.42            |

---

> ### Author Response · Authors · 2024-11-20
> **Supplement to the Response for Point 1**
>
> ***Performance of Proposed Hybrid Multi-Defense Prompt and Mainstream Prompting Methods on 10-shot Node Classification and Link Prediction Tasks.***
>
>
> | Methods               | Node (Cora) | Link (Cora) | Node (Citeseer) | Link (Citeseer) | Node (Pubmed) | Link (Pubmed) | Node (Wisconsin) | Link (Wisconsin) | Node (ogbn-arxiv) | Link (ogbn-arxiv) |
> |---------------------------|-----------|-----------|---------------|---------------|-------------|-------------|----------------|----------------|-----------------|-----------------|
> | Pre-train & Fine-tune      | 52.23     | 67.55     | 58.82         | 60.57         | 50.35       | 59.23       | 29.12          | 65.49          | 14.38           | 55.41           |
> | GPPT                       | 47.77     | 65.47     | 54.09         | 74.48         | 48.91       | 55.35       | 22.43          | 59.13          | 7.74            | 52.86           |
> | ProG                       | 65.90     | **77.23** | 53.24         | 65.95         | 49.33       | 74.03       | 41.76          | **82.35**      | 5.25            | 80.50           |
> | Gprompt                    | 70.81     | 66.45     | 76.46         | 50.46         | 60.10       | 61.58       | **43.96**      | 50.43          | **24.31**       | 75.55           |
> | GPF                         | 69.93     | 73.35     | 53.82         | 60.37         | 39.26       | 50.08       | 31.72          | 57.75          | 10.36           | 63.03           |
> | GPF-plus                   | 72.79     | 88.24     | **78.46**     | 73.02         | 42.45       | **88.75**   | 32.26          | 59.57          | 14.12           | **81.40**       |
> | *MD-PT                     | **73.45** | 74.55     | 75.85         | **77.49**     | **62.34**   | 62.98       | 35.65          | 54.42          | 20.45           | 68.45           |
>
>
> ***Performance of Proposed Hybrid Multi-Defense Prompt and Mainstream Prompting Methods on Graph Classification Tasks.***
>
> | Methods \ Datasets         | MUTAG 5-shot | MUTAG 10-shot | COLLAB 5-shot | COLLAB 10-shot | PROTEINS 5-shot | PROTEINS 10-shot | IMDB-B 5-shot | IMDB-B 10-shot |
> |----------------------------|--------------|---------------|---------------|----------------|-----------------|------------------|----------------|----------------|
> | Pre-train & Fine-tune   | 68.00        | 68.00         | 61.28         | 62.77          | 53.37           | 65.51            | 63.00          | 67.00          |
> | GPPT                    | 64.00        | 68.67         | 50.48         | 51.97          | 59.55           | 60.33            | 51.25          | 54.48          |
> | ProG                    | 70.00        | 66.67         | 63.60         | 54.70          | 62.92           | **67.22**        | 61.12          | 66.12          |
> | Gprompt                 | 72.67        | 70.67         | 64.53         | 64.25          | 61.24           | 58.65            | 60.50          | **69.50**      |
> | GPF                     | 64.67        | 65.33         | 61.80         | 62.45          | 64.61           | 66.29            | 60.17          | 66.75          |
> | GPF-plus                | 68.00        | 72.00         | 62.20         | 63.75          | 65.17           | 63.26            | 60.75          | 67.12          |
> | *MD-PT                   | **74.67**    | **74.35**     | **65.07**     | **65.63**      | **67.39**       | 66.52            | **63.13**      | 62.57          |
>
>
>
> Through the addition of these experiments, we find that the auxiliary robustness system we design does not sacrifice accuracy in favor of robustness. In fact, it maintains strong competitiveness on many task datasets, which is a pleasant surprise during the expansion of the experiments. : ) ! We speculate that the original datasets might already contain some inherent noise, and our robustness enhancement tool further purifies the data during the training process.
>
> ( By the way, some of the data in our experimental results are referenced from the recent benchmark work [3] presented at NeurIPS 2024. Thank you for your recommendation!  it has been a great help to us within the limited time frame. )
>
> We sincerely hope that the additional data will meet your approval. If there are any further points of concern, please feel free to let us know!
>
> [3] ProG: A Graph Prompt Learning Benchmark

---

> ### Author Response · Authors · 2024-11-20
> **Response to Point 2**
>
> Thank you for your valuable feedback. We understand your concerns regarding the theoretical analysis in Section 4.3. The purpose of this section is, on one hand, to provide a foundational and formal understanding of the robustness prompt we propose, particularly in adversarial settings. On the other hand, we draw from the existence proof in GPF [4] and extend it to biased attack scenarios for further exploration, providing important support for the existence of our designed robust prompt tuning method. This is the key point we aim to highlight in our theoretical proof. The existence of the method is often of greater significance, as it establishes the theoretical foundation for its reliability and robustness. As for how prompt tuning enhances robustness, the results from a large number of experiments may provide a more intuitive and convincing explanation.   : ) !   We hope that this section will not affect your overall understanding and judgment of the contribution and completeness of the paper.   : ) !
>
> [4] Universal prompt tuning for graph neural networks.

---

> ### Author Response · Authors · 2024-11-20
> **Response to Point 3**
>
> Thank you very much for your valuable suggestion! It has helped us evaluate and validate the proposed prompt more comprehensively. We have added this experimental result to the ablation study in the revised version, and the updated coarse-grained data in the table is shown below. However, this result is already reflected in the experiments of the paper. The situation you mentioned is actually the method used in GPF/GPF-plus. Their approach is to apply a universal prompt to all nodes for tuning.
>
> ***Therefore, we can observe an interesting point: when our method does not use the attention mechanism, these two methods are essentially a special case of our approach! ! !***
>
> As for which types of the nodes should be given more weights in the robust tuning process, the main focus of this work is on node-specific prompts, where the weights are discussed in terms of how a node integrates different prompts it possesses. The selection of nodes based on specific weights is not within the scope of this paper. Therefore, during the prompt tuning phase, the focus is primarily on tuning the prompt function and the attention module, rather than selecting the nodes. I hope this explanation is clear and understandable to you. : ）!
>
>
> | Ablation Variants                 | Cora 5-shot Meta 0.25 | Cora 5-shot DICE 0.5 | Citeseer 5-shot Meta 0.25 | Citeseer 5-shot DICE 0.5 | Cora 10-shot Meta 0.25 | Cora 10-shot DICE 0.5 | Citeseer 10-shot Meta 0.25 | Citeseer 10-shot DICE 0.5 |
> |-----------------------------------|-----------------------|----------------------|---------------------------|-------------------------|------------------------|-----------------------|----------------------------|---------------------------|
> | MD-PT with all Prompts            | 60.12%                | 54.55%               | **64.20%**                | 52.39%                  | **71.02%**             | **62.30%**            | **67.04%**                | 55.35%                    |
> | MD-PT w/o $p_{degree}$            | **62.31%**            | 53.05%               | 63.68%                    | 51.76%                  | 67.87%                 | 61.36%                | 66.67%                    | **56.31%**                |
> | MD-PT w/o $p_{sim}$               | 58.46%                | 52.02%               | 58.97%                    | **53.21%**              | 69.92%                 | 59.48%                | 61.57%                    | 54.39%                    |
> | MD-PT w/o $p_{detector}$          | 59.32%                | 55.37%               | 60.42%                    | 51.98%                  | 69.09%                 | 60.99%                | 65.69%                    | 55.18%                    |
> | MD-PT w/o $\mathcal{L}_{kl}$      | 59.86%                | 55.28%               | 60.42%                    | 52.14%                  | 69.83%                 | 60.81%                | 64.42%                    | 54.23%                    |
> | MD-PT w/o $\mathcal{L}_{o}$       | 59.59%                | 51.20%               | 59.94%                    | 50.00%                  | 69.23%                 | 58.33%                | 65.64%                    | 55.01%                    |
> | MD-PT w/o $\mathcal{L}_{s}$       | 58.32%                | **56.46%**           | 60.84%                    | 51.50%                  | 69.87%                 | 60.72%                | 66.34%                    | 54.58%                    |

---

> ### Author Response · Authors · 2024-11-20
> **Response to Point 4**
>
> Thank you for your careful reading and valuable suggestions regarding the content and flow of our paper! Due to space limitations, we were unable to provide a detailed introduction of this section in the main text. However, we have provided a more thorough explanation in the revised version. This content is intended more for the appendix, and we hope it hasn't affected your overall assessment of the methods in our paper. We will first provide the supplementary content below to assist with your understanding, which will be included in the revised version shortly.
>
>
> Adaptive attacks are a proposed new robustness evaluation standard and are a type of white-box attack. They indicate that, with a thorough understanding of the defense model, attackers can bypass defenses using corresponding strategies. Thus, existing defenses are not as robust as they were evaluated in their papers. These defenses are vulnerable to adaptive attacks because most of them are designed based on some specific properties that can be used to differentiate adversarial edges from original edges. Adversarial
> modifications on graphs often violate some intrinsic properties shared by the real-world graphs. The adversary can easily defeat the defenses by imposing constraints on the same properties during the attack. Therefore, the key to enhancing adaptive robustness is not relying on artificially defined properties. To overcome this dependence, [5] proposed a new approach from the out-of-distribution perspective, arguing that adversarial edges generated through attacks are  inherently out-of-distribution compared to the original edges. Thus, by modeling the entire out-of-distribution problem, an integrated out-of-distribution detector is trained using adversarially generated edges to detect and remove perturbations.
>
>
> By detecting these perturbation edges, the nodes at both ends of the edges become the focal points of the attack method. This is precisely why we add prompts to these nodes. This aligns well with our motivation: these vulnerable nodes are the ones where we should add prompts. The purpose of adding prompts is to help these weak nodes have effective information during the task, rather than providing misleading information.
>
>
>
> We hope this supplementary response meets your satisfaction. : )
>
>
> [5] BOOSTING THE ADVERSARIAL ROBUSTNESS OF GRAPH NEURAL NETWORKS: AN OOD PERSPECTIVE

---

> ### Author Response · Authors · 2024-11-20
> **Response to Point 5**
>
> This is a very detailed question, and your thorough reading has provided us with significant insights. In this work, $\tau_{degree}$, $\tau_{sim}$, and $\tau_{tune}$ are empirically set as tunable hyperparameters. Adjusting them does not require extensive expert knowledge, a basic understanding of graph properties is sufficient, and the process is highly convenient. Designing them to be adaptive is an intriguing idea, however, it seems quite challenging to implement on graphs. In my opinion, creating adaptive values might require more expert knowledge to be incorporated into the prompt-tuning process. Additionally, adapting to various types of graphs poses a significant challenge, which can be further explored in future work. : )
>
>
> Regarding Eq. 15, we perform further fine-grained pruning of the graph based on the features after adding the prompts. From an experimental perspective, such pruning is both necessary and meaningful in attack scenarios. This pruning helps to eliminate factors that are irrelevant or even harmful to the training process. Especially in biased graph environments, information redundancy is not beneficial. On the contrary, the precision and effectiveness of the information are what should be pursued. By setting appropriate thresholds, we can make the GPL more robust and stable during the tuning process.
>
>
> We hope our response is satisfactory and meets your expectations. : )

---

> ### Author Response · Authors · 2024-11-20
> **Response to Point 6**
>
> This is an excellent experimental setup. Thank you so much for providing us with such an interesting transfer scenario! It allows us to explore the potential of our prompt from a completely new perspective!
>
>
> We explore prompt tuning under 5-shot conditions on datasets Cora, Citeseer, and Cora_ml with one type of attack and evaluate the performance on the other three attack datasets. We apply our two tools to GPF/GPF-plus to observe robustness performance, and the coarse-grained results are shown in the table below. More refined table data will be presented in the updated version.
>
> As seen, our IA-PT method still demonstrates excellent robustness, while MD-PT is slightly less effective, but still consistently improves the robustness of GPF/GPF-plus. This aligns well with what we describe in our paper: avoiding difficulties is often much easier than facing them head-on.
>
>
> This is a highly creative transfer experiment. Thank you for your guidance, and we hope that our experimental results and responses meet your expectations. We also sincerely hope to receive your recognition!
>
>
> - Cora
>
>
> | Prompts   | Attacks | M-0.25 | D-0.5  | R-0.5  | H-0.5  |
> |-----------|---------|--------|--------|--------|--------|
> | GPF   | M-0.25  | 31.44  | 32.47  | 20.23  | 26.26  |
> |           | D-0.5   | 33.97  | 23.38  | 28.21  | 32.70  |
> |           | R-0.5   | 36.92  | 36.37  | 36.24  | 34.37  |
> |           | H-0.5   | 29.16  | 35.85  | 29.30  | 29.02  |
> | GPF-plus | M-0.25 | 35.56  | 35.51  | 34.92  | 36.33  |
> |           | D-0.5   | 37.19  | 36.87  | 38.19  | 37.10  |
> |           | R-0.5   | 42.26  | 40.35  | 40.26  | 44.35  |
> |           | H-0.5   | 33.56  | 30.79  | 35.92  | 32.79  |
> | *MD-PT | M-0.25  | 47.06  | 42.40  | 43.11  | 47.89  |
> |           | D-0.5   | 47.08  | 40.15  | 42.81  | 47.13  |
> |           | R-0.5   | 48.04  | 45.82  | 50.34  | 53.52  |
> |           | H-0.5   | 50.15  | 41.45  | 44.67  | 47.48  |
> | *IA-PT | M-0.25  | 65.22  | 49.50  | 60.16  | 62.53  |
> |           | D-0.5   | 64.54  | 52.24  | 57.60  | 61.57  |
> |           | R-0.5   | 57.85  | 48.04  | 54.69  | 60.46  |
> |           | H-0.5   | 60.41  | 53.62  | 55.13  | 62.55  |
>
>
>
> - Citeseer
>
>
> | Prompts   | Attacks | M-0.25 | D-0.5  | R-0.5  | H-0.5  |
> |-----------|---------|--------|--------|--------|--------|
> | GPF   | M-0.25  | 20.26  | 20.85  | 32.80  | 35.79  |
> |           | D-0.5   | 36.22  | 25.37  | 25.37  | 20.75  |
> |           | R-0.5   | 25.37  | 21.96  | 21.96  | 21.96  |
> |           | H-0.5   | 31.26  | 25.37  | 20.15  | 19.86  |
> | GPF-plus | M-0.25 | 25.37  | 24.46  | 38.76  | 40.48  |
> |           | D-0.5   | 26.98  | 28.77  | 29.49  | 27.40  |
> |           | R-0.5   | 34.29  | 35.71  | 35.34  | 35.68  |
> |           | H-0.5   | 20.20  | 33.12  | 34.78  | 33.07  |
> | *MD-PT | M-0.25  | 42.81  | 39.62  | 40.46  | 44.85  |
> |           | D-0.5   | 42.18  | 36.72  | 36.32  | 38.88  |
> |           | R-0.5   | 40.46  | 37.82  | 37.77  | 38.75  |
> |           | H-0.5   | 41.82  | 38.39  | 35.55  | 41.08  |
> | *IA-PT | M-0.25  | 55.66  | 48.76  | 55.04  | 58.35  |
> |           | D-0.5   | 57.41  | 55.34  | 51.24  | 53.61  |
> |           | R-0.5   | 56.52  | 54.03  | 48.96  | 54.21  |
> |           | H-0.5   | 57.94  | 52.73  | 50.47  | 58.55  |
>
>
> - Cora_ml
>
>
> | Prompts   | Attacks | M-0.25 | D-0.5  | R-0.5  | H-0.5  |
> |-----------|---------|--------|--------|--------|--------|
> | GPF   | M-0.25  | 28.99  | 32.07  | 33.07  | 30.42  |
> |           | D-0.5   | 36.75  | 33.79  | 34.79  | 31.71  |
> |           | R-0.5   | 34.63  | 34.47  | 39.31  | 39.95  |
> |           | H-0.5   | 31.39  | 21.74  | 34.19  | 32.54  |
> | GPF-plus | M-0.25 | 33.15  | 35.67  | 35.15  | 36.75  |
> |           | D-0.5   | 34.35  | 33.99  | 32.31  | 32.03  |
> |           | R-0.5   | 38.11  | 37.67  | 36.03  | 37.83  |
> |           | H-0.5   | 40.03  | 40.45  | 37.11  | 39.95  |
> | *MD-PT | M-0.25  | 53.82  | 43.57  | 42.17  | 47.51  |
> |           | D-0.5   | 43.44  | 44.55  | 40.02  | 51.25  |
> |           | R-0.5   | 52.76  | 41.57  | 45.24  | 50.80  |
> |           | H-0.5   | 41.35  | 41.34  | 39.45  | 52.36  |
> | *IA-PT | M-0.25  | 61.61  | 54.89  | 59.96  | 65.57  |
> |           | D-0.5   | 58.19  | 52.52  | 53.02  | 57.56  |
> |           | R-0.5   | 62.19  | 49.13  | 54.00  | 61.92  |
> |           | H-0.5   | 62.10  | 52.71  | 59.48  | 58.77  |

---

> ### Author Response · Authors · 2024-11-23
> **Looking Forward to Your Reply**
>
> Dear Reviewer zGFc,
>
> Among so many submissions, we feel truly fortunate to have the opportunity to discuss with you. We fully understand that you have many other concurrent tasks and greatly cherish this opportunity to communicate with you.
>
> ***As the time window for the rebuttal is closing soon, and in light of the relatively positive feedback from other reviewers, your feedback is crucial for the further development of our paper! : )*** ! We have made several improvements to the manuscript based on your valuable suggestions, and these revisions would greatly benefit from your further input and approval. We greatly appreciate your dedication to enriching our work.
>
> ***We are extremely eager for the revised manuscript to have the opportunity to be reviewed by you, we would greatly appreciate knowing if there are any additional questions we can answer to help raise the score! : )***
>
> We are warmly looking forward to your response and we are glad to discuss further with you !
>
> Best,
>
> Authors of Submission 6211

---

> ### Author Response · Authors · 2024-11-26
> **A Gentle Reminder to Reviewer zGFc**
>
> Dear Reviewer zGFc,
>
> ***We truly, truly understand that this is an exceptionally challenging and busy period.*** We deeply appreciate the time and effort you have invested in reviewing our work. We have worked hard to carefully address the concerns raised in your review and have substantially revised our manuscript accordingly.
>
> As this period is approaching its end on ***November 27***, we kindly ask if you could please let us know whether our revisions have satisfactorily resolved your concerns or if there are any remaining issues or questions you would like us to address before the rebuttal deadline. ***As we are unable to make further modifications to the manuscript after November 27, we will do our utmost to ensure it meets your high standards.*** We would greatly appreciate it if you could reevaluate our paper based on the revisions and consider updating your evaluation accordingly. Your feedback is invaluable to us and plays a crucial role in enhancing the quality of our work.
>
> Thank you for your consideration and we are looking forward to your replies!
>
> Yours sincerely,
>
> Authors of Submission 6211

---

> ### Author Response · Authors · 2024-11-28
> **Friendly reminder to Reviewer zGFc**
>
> Dear Reviewer zGFc,
>
> This is a friendly reminder. : )  You have provided us with many valuable suggestions to enhance our paper, all of which we have addressed one by one.  We sincerely hope that our revised manuscript will meet your expectations.
>
> To the best of our knowledge, we are the first to explore robust tuning in the field of graph prompt learning. Therefore, we sincerely hope you can provide us with further insightful guidance and consider reevaluating our work.
>
> We are warmly looking forward to hearing the good news from you ! Please do not hesitate to give us any further feedback at your earliest convenience.
>
> Best,
>
> Authors of Submission 6211

---

> ### Author Response · Authors · 2024-11-29
> **Warm Thanksgiving Wishes ！**
>
> Dear Reviewer zGFc,
>
> ***Happy Thanksgiving! Today is a perfect day to express our heartfelt gratitude. We sincerely appreciate the time and effort all reviewers have dedicated to our work. I hope this message finds you well and wish you a wonderful Thanksgiving filled with warmth and joy.  : )***
>
> If our previous responses have effectively addressed all of your concerns, it would make our day even more meaningful. ***We sincerely hope to receive your positive feedback on this perfect day to help us shape a more perfect piece of work. : )***
>
> Best Wishes,
>
> Authors of Submission 6211

---

### Official Review · Reviewer_1kqh · 2024-11-03

**Soundness:** 3
**Presentation:** 3
**Contribution:** 3
**Rating:** 6
**Confidence:** 3

**Summary:**

This paper explores enhancing the robustness of graph prompt-tuning methods against adversarial attacks in graph neural networks (GNNs). It identifies a vulnerability in current prompt-based methods, which are highly susceptible to adversarial alterations. The proposed solution is a "shield defense system" that enhances robustness through two strategies: Direct Handling and Indirect Amplification. Direct Handling customizes multi-defense prompts based on node-specific attributes, targeting biased nodes introduced by attacks, while Indirect Amplification leverages few-shot learning and selectively focuses on untainted information, thereby circumventing the misleading data introduced by adversaries. Theoretical validation demonstrates that this approach can maintain unbiased outputs despite adversarial perturbations. Experimental results show that the defense system enhances resilience under various attack scenarios, outperforming existing prompts in maintaining accuracy and robustness.

**Strengths:**

The idea and structure is easy to follow.

**Weaknesses:**

1. Some of the tables are confusing, for example, the table 1. Pre-Training Strategies & Prompts appear simultaneously in the top left corner. Should we design the table in a better way or use other charts to represent these results?
2. Section 2 Related work, if there is no more subsection 2.1, is not necessary.
3. The whole framework should have a figure to clearly show the stages of prompt design and robust optimization strategies

**Questions:**

1. Why other methods' results are getting close to the proposed methods in 10-shot experiments? Is it possible that other methods will outperform the proposed methods with even more shots?
2. Adding the figures to represent prompt design and robust optimization strategies stages could be better.

---

> ### Author Response · Authors · 2024-11-21
> **Thanks to the reviewer and the Response**
>
> Dear Reviewer 1kqh，
>
> Thank you very much for the dedicated review of our paper. We are fully aware of the commitment and time your review entails. Your efforts are deeply valued by us.
>
>
> Thank you very much for your positive evaluation of our paper; it is a great encouragement to us. Regarding the concerns you raised, we hope the following responses will address them satisfactorily and lead to further recognition of our work.  : )
>
> - Thank you very much for your attention to the presentation of our Tables. Regarding the strategies in Table 1, we have actually provided a detailed explanation in the paper. : )  Since the prompt function is a method applied to the input data while keeping the pre-trained model fixed, it is more intuitive to explore different prompt methods under the same pre-trained model. ***This is also reflected in the setup of Table 1, where "Pre-Training Strategies" and "Prompts" are presented as separate columns rather than as a combined relationship like "Pre-Training Strategies & Prompts".***  Specifically, GraphMAE and GraphCL are part of "Pre-Training Strategies", while ProG, GPF, and others belong to  "Prompts" . Since the prompt functions in GraphPrompt and MultiGPrompt rely on their respective customized pre-training strategies, we did not separate them in order to ensure a fair comparison. : ) !  Thank you for your careful consideration. We will make further optimizations to improve readability for the readers.
>
> - Regarding the "Related Work" section, we have updated the section numbering as per your request, and it will be reflected in the revised version of our paper shortly!  : )
>
> - As for the experimental results under the 10-shot condition, your observation is very insightful, and this is actually a normal phenomenon. Typically, we assume that as the training data increases, the model is able to learn more knowledge. Since attacks often aim to achieve greater attack effectiveness with fewer operations, an increase in data might lead to the model learning more positive information. This can result in the dilution of negative information caused by attacks in the training set, thereby narrowing the gap in robustness performance.  ***However, this phenomenon is not absolute. If the additional data contains more negative information, we are confident that our method will still maintain a more significant robustness performance.*** Moreover, mainstream prompt methods, including GPF/GPT-plus, can be considered as special cases within our prompt system, representing a subset relationship. Therefore, regardless of the settings, our method is at least capable of demonstrating performance comparable to that of mainstream methods. : ) ! ***At the same time, in a recent benchmark study [1] that has just been accepted to NIPS 2024, the experimental results also show that larger shot sizes do not necessarily lead to better performance than smaller ones. The key factor determining prompt performance actually lies in the selection of shot nodes. Therefore, in response to the issue you raised, in some cases with larger shot sizes, such as 10-shot or beyond, our method may exhibit even more significant robustness compared to the smaller shot results.
>
>
> - Thank you very much for your attention to the figures and workflow in our paper. We greatly value your reading experience, and the revised version of our paper will be updated shortly : ) !
>
>
>
> [1] ProG: A Graph Prompt Learning Benchmark
>
> （Additionally, we have included many interesting experiments in the response to other reviewers. If you're interested, feel free to take a look and discuss with us. We would greatly appreciate your involvement and feedback !）
>
> ***We sincerely hope that our responses meet your expectations. If you have any other concerns, please do not hesitate to let us know. We will do our best to provide a detailed explanation and engage in further communication. As the rebuttal period is very tight, we sincerely hope to gain your further recognition within the limited time available : ) !***

---

> ### Author Response · Authors · 2024-11-23
> **Kindly Reminder : )**
>
> Dear Reviewer 1kqh，
>
> As the time window for the rebuttal is closing soon, we are truly grateful for your positive attitude toward our paper! : )  Among so many submissions, we feel truly fortunate to have the opportunity to discuss with you. We fully understand that you have many other concurrent tasks and greatly cherish this opportunity to communicate with you.
>
> ***If our revised manuscript still has the opportunity to be reviewed by you, we would greatly appreciate knowing if there are any additional questions we can answer to help raise the score! : )***
>
> Best,
>
> Authors of Submission 6211

---

> > ### Comment · Reviewer_1kqh · 2024-11-26
> >
> > Thank authors reply, I will maintain my score. The results partially reflect my question, the authors should provide more details.

---

> > > ### Author Response · Authors · 2024-11-26
> > >
> > > We sincerely appreciate your positive feedback and support. : )  Thank you once again for your time and effort in reviewing our paper ! Of course! If there are any details that could help us earn a higher score from you, please don’t hesitate to let us know!  : )

---

### Official Review · Reviewer_iA8J · 2024-11-04

**Soundness:** 3
**Presentation:** 2
**Contribution:** 3
**Rating:** 6
**Confidence:** 3

**Summary:**

This paper introduces a novel approach to graph prompt tuning focused on robustness in adversarial attack scenarios. It proposes a highly extensible shield defense system with a hybrid multi-defense prompt and robust prompt tuning strategy, demonstrating theoretical feasibility. Extensive experiments in few-shot scenarios under various adversarial attacks show that their strategies significantly enhance prompt tuning resilience in downstream biased tasks. The use of feature-based prompts allows real-time adjustments, and the paper highlights the effectiveness of their methods in both adaptive and non-adaptive attack scenarios.

**Strengths:**

The proposed mixed multi-layer defense strategy in this paper demonstrates technical innovation by combining various defense mechanisms, such as feature-based prompting and real-time adjustments. This approach significantly enhances the system’s tolerance to adversarial attacks, offering a fresh perspective on the research of graphic prompting adjustment systems.

**Weaknesses:**

One weakness of the paper is the absence of a comparative analysis with existing systems or defense strategies against adversarial attacks.

The paper primarily focuses on theoretical analysis and experimental validation in controlled settings. However, the lack of real-world implementation or case studies hinders the practical applicability assessment of the proposed method. To enhance the relevance and impact of the research, future work could involve implementing the system in real-world scenarios and evaluating its performance in practical applications.

The evaluation section of the paper might benefit from a more comprehensive selection of evaluation metrics to assess the proposed method’s performance accurately. Specifically, incorporating additional metrics such as computational efficiency, scalability, or user experience aspects could provide a more holistic evaluation of the system’s capabilities and limitations. Including a diverse set of evaluation criteria would offer a more thorough understanding of the method’s strengths and weaknesses.

The paper could further strengthen its impact by discussing the generalizability of the proposed method to different types of adversarial attacks or diverse datasets. Providing insights into how the approach could be adapted or extended to address a broader range of security challenges would enhance the paper’s contribution to the field.

**Questions:**

Refer to the weaknesses above.

---

> ### Author Response · Authors · 2024-11-20
> **Thanks to the reviewer and the Response**
>
> Dear Reviewer iA8J，
>
> Thank you very much for the dedicated review of our paper. We are fully aware of the commitment and time your review entails. Your efforts are deeply valued by us.
>
> Thank you very much for providing us with many directions for future work and different evaluation perspectives. Your suggestions have been very insightful and beneficial to us. Regarding the issues you raised, I hope that the new additions we have made will meet your expectations. : )
>
> - Since the field of graph prompt learning is still in its early stages of rapid development, and there has been almost no work focusing on robustness in graph prompt learning, it has been challenging to find a fair comparison for our defense system. We considered directly comparing it with some robust graph neural networks, but the application scenarios differ, and robustness at the model level and robustness in the prompts are two relatively distinct concepts. On the other hand, we also considered designing robust variants of pre-trained models, but we found that these robust variants are still comparisons at the model level, whereas our pre-trained models remain frozen throughout the prompt tuning phase. Therefore, these comparison methods were not entirely fair. As a newly proposed concept for prompt robustness, the best way to validate its robustness is to apply it to existing prompts and observe how they perform under biased attacked data. Through extensive experimental feedback, we aim to provide a comprehensive evaluation of our system. Of course, we hope that more work focusing on improving the robustness of graph prompts will emerge in the future. We will stay updated on these developments and conduct comprehensive comparisons in our future work. :) !
>
> - Regarding the evaluation on more datasets that you mentioned, we have added a significant number of experiments. We extended the evaluation of our system to cover all ***graph-related tasks, including node classification, link prediction, and graph classification***, greatly expanding the scope of our application scenarios. For node classification and link prediction tasks, we select homophilic datasets (Cora,Citeseer,Pubmed) and heterophilic datasets (Wisconsin) and explore the prompts on large graph (ogbn-arxiv). We conduct experiments under 5-shot and 10-shot settings, where the link prediction task includes 5,000 positive edges and 5,000 negative edges. For the graph classification task, in order to be more comprehensive, we chose the molecular dataset MUTAG, the social network dataset COLLAB, the protein dataset PROTEINS, and the social network dataset IMDB-BINARY.  We also conduct experiments under 5-shot and 10-shot settings. We presented the extensive supplementary experiments in the response boxes of other reviewers, evaluating our system from various aspects. It can be seen that while considering robustness, our system maintains a balance with accuracy across many tasks. :)
>
> - In response to your concerns about scalability and transferability, ***we also conducted transfer experiments under different attack conditions in the response boxes for other reviewers, demonstrating the strong transferability and scalability of our system.***
>
>
> ***We hope our response meets your satisfaction and further acknowledges our work. If you have any other concerns, feel free to let us know at any time. We will do our best to address every question and need you may have. : ) !***
>
> (All the datasets information，additional experimental data and more detailed tables will be updated in our revised version shortly.   : )！ )

---

> ### Author Response · Authors · 2024-11-23
> **Kindly Reminder  : )**
>
> Dear Reviewer iA8J,
>
> As the time window for the rebuttal is closing soon, we are truly grateful for your positive attitude toward our paper! : )  Among so many submissions, we feel truly fortunate to have the opportunity to discuss with you. We fully understand that you have many other concurrent tasks and greatly cherish this opportunity to communicate with you.
>
> ***If our revised manuscript still has the opportunity to be reviewed by you, we would greatly appreciate knowing if there are any additional questions we can answer to help raise the score! : )***
>
> Best,
>
> Authors of Submission 6211

---

### Author Response · Authors · 2024-11-21
**Summary of Rebuttal Revision**

Dear Reviewers,

We sincerely thank all the reviewers for their patient and thoughtful evaluation of our work. In response to the valuable feedback, we have uploaded an updated version of the manuscript, which includes several major revisions. The details are as follows:

- We have added a detailed description in **A.2** discussing adaptive robustness from the perspective of out-of-distribution generalization.
- We have expanded the description of all datasets for node classification, link prediction, and graph classification tasks in **A.4.1**.
- We have added the performance of the prompt functions on node classification and link prediction tasks in **A.8**, and the performance on graph classification tasks in **A.9**
- we have added a transfer attack experiment in  **A.10** to validate the scalability and transferability of our system.
- We have added the ablation experiment of "MD-PT with all Prompts" in **Table 3**.
- We have made changes to the section numbering and revised the grammatical details throughout the manuscript.

We have highlighted all the modifications and additions throughout the manuscript in  $\textcolor{red}{red \ part}$. We hope that the revised manuscript can help address the concerns and resolve the issues raised by the reviewers.


As the rebuttal period is coming to a close, we sincerely hope to receive your feedback. We are committed to addressing all of your concerns to the best of our ability.

Best,

Authors of Submission 6211

---

### Meta-Review · Area_Chair_WL6h · 2024-12-19

**Metareview:**

The paper proposes a defense method to enhance the robustness of graph prompt tuning against adversarial attacks. While the proposed method is interesting and offers a new perspective on the graph prompt defense, the paper has some concerns. First, the paper is only limited to the node classification tasks (several other tasks have been added in the rebuttal). Second, although some theoretical analyses have been conducted, the analysis is not well-connected with the proposed method. Third, the paper needs to be polished in several places. Last, it lacks discussion and comparison with some previous studies. Please make a careful revision.

**Additional Comments On Reviewer Discussion:**

The author addresses the reviewer's concern by adding experiments on different tasks and explaining the intuition. However, some concerns are not fully addressed such as the theoretical analysis connection with the proposed method and some defense baselines, etc.

---

### Decision · Program_Chairs · 2025-01-22

Accept (Poster)